# Achieving Olympia-Level Geometry Large Language Model Agent via Complexity Boosting Reinforcement Learning

**Haiteng Zhao**[*,1]**, Junhao Shen**[*,1,2]**, Yiming Zhang**[*,1]**, Songyang Gao**[1]**, Kuikun Liu**[1]
**Tianyou Ma**[1,3]**, Fan Zheng**[4]**, Dahua Lin**[1,5]**, Wenwei Zhang**[1,†]**, Kai Chen**[1,†]
[1]Shanghai AI Laboratory   [2]Shanghai Jiao Tong University   [3]Peking University
[4]ICMAT, Spanish National Research Council   [5]MMLab, The Chinese University of Hong Kong
{zhaohaiteng,zhangwenwei,chenkai}@pjlab.org.cn

## Abstract

Large language model (LLM) agents exhibit strong mathematical problem-solving abilities and can even solve International Mathematical Olympiad (IMO) level problems with the assistance of formal proof systems. However, due to weak heuristics for auxiliary constructions, AI for geometry problem solving remains dominated by expert models such as AlphaGeometry 2, which rely heavily on large-scale data synthesis and search for both training and evaluation. In this work, we make the first attempt to build a medalist-level LLM agent for geometry and present InternGeometry. InternGeometry overcomes the heuristic limitations in geometry by iteratively proposing propositions and auxiliary constructions, verifying them with a symbolic engine, and reflecting on the engine's feedback to guide subsequent proposals. A dynamic memory mechanism enables InternGeometry to conduct more than two hundred interactions with the symbolic engine per problem. To further accelerate learning, we introduce Complexity-Boosting Reinforcement Learning (CBRL), which gradually increases the complexity of synthesized problems across training stages. Built on InternThinker-32B, InternGeometry solves 44 of 50 IMO geometry problems (2000-2024), exceeding the average gold medalist score (40.9), using only 13K training examples, just $0.004\%$ of the data used by AlphaGeometry 2, demonstrating the potential of LLM agents on expert-level geometry tasks. InternGeometry can also propose novel auxiliary constructions for IMO problems that do not appear in human solutions.

## 1 Introduction

Large language model (LLM) agents have demonstrated general problem-solving ability across domains such as mathematics and programming. By interacting with tools such as code interpreters and LEAN (Moura & Ullrich, 2021) multiple times, LLM agents can reason and reflect on the tool execution feedback and progressively solve complex problems. Such a universal paradigm can obtain medalist-level performance on International Mathematical Olympiad (IMO) level problems and is believed to have better generalization ability (Huang & Yang, 2025; Luong & Lockhart, 2025).

However, when faced with geometry problems, the potential of LLM agents is still underexplored. The IMO-level geometry problems usually require extra-long proving steps, whose solution not only combines various geometry theorems but also contains creative auxiliary constructions that have weak heuristics and require multiple trials, as shown in Figure 1. Consequently, current state-of-the-art approaches (Trinh & Luong, 2024; Chervonyi et al., 2025; Chen et al., 2025a) mainly adopt expert models learned from large-scale synthesized data to guide the large-scale search with a symbolic engine for finding the geometry proof. Given the success of LLM agents in other mathematical domains, this raises a natural question: *Can we adopt LLM agents in solving geometry problems for achieving higher efficiency and generalization?*

To answer this question, this paper makes the first attempt to investigate LLM agents for solving IMO-level geometry problems and proposes InternGeometry, an LLM agent that obtains medalist-

**(a) Original Geometry Sutrcutre in IMO 2018 P6**

**(b) Auxiliary Constructions Proposed by InternGeometry**

Figure 1: **An example of an IMO-level geometry problem.** (a) The configuration in IMO 2018 Problem 6 appears simple, but its solution relies on sophisticated constructions, as illustrated in (b).

level performance for solving geometry problems. We first identify the limitations in the current open-sourced geometry symbolic engines and build InternGeometry-DDAR, whose expression capacity covering most of the IMO geometry problems. Taking InternGeometry-DDAR as a tool, InternGeometry solves the geometry problem through long-term LLM-tool interactions, where the LLMs continuously propose propositions or auxiliary constructions after thinking with natural language, verify those ideas in the symbolic engine with formal language, and reflect on the feedback from the symbolic engine at each interaction. In the long-term reasoning process, InternGeometry adopts dynamic memory to maintain the exploration history and the observed geometry properties in a compact form, which not only reduces the context without losing key information but also guides diverse explorations in future interactions. By extra-long-horizon LLM-tool interactions with memory, InternGeometry can conquer the weak heuristics of geometry proof and progressively find the feasible solution based on the accumulated geometry properties observed during explorations. Such a design also aligns with human experts who obtain insights into auxiliary construction by exploratory probing (Trinh & Luong, 2024; Chervonyi et al., 2025).

To train InternGeometry, we first apply cold start training using 7K examples created by formalizing existing geometry problems and constructing trajectory data. After the cold start, we introduce a complexity-boosting reinforcement learning (CBRL) framework, a multi-stage curriculum RL pipeline (Wang et al., 2025b; Chen et al., 2025b; Zhang et al., 2025b; Parashar et al., 2025), to further improve training efficiency. Specifically, we build a data synthesis pipeline that can generate geometry tasks with a specified complexity (e.g., required proof steps). At each stage, we first synthesize problems at the current complexity level, then perform RL training on the current InternGeometry model, and update the target complexity based on the results to best fit the current model. Over iterations, the synthesized problems become highly challenging, providing the foundation for acquiring expert-level capabilities on high-difficulty tasks.

We conduct extensive experiments to verify the effectiveness of InternGeometry. InternGeometry solves 44 out of 50 geometry problems from 2000 to 2024, surpassing the average score of IMO gold medalists (40.9 points) and the score of AlphaGeometry2 (42) and SeedGeometry (43), and it also solves the geometry problem in IMO 2025. Notably, the model attains this performance with only approximately 13K training examples, 0.004% of AlphaGeometry2 and 0.006% of SeedGeometry. Our ablation studies further demonstrate that long-horizon proof interaction is critical to the agent's ability: removing proposition proving steps and only allowing the agent to add auxiliary constructions significantly degrades performance, substantiating the importance of long-horizon trial-and-error for the weak-to-strong heuristic transition. Complexity escalation plays a pivotal role in RL convergence: directly training on high-difficulty data leads to low task completion rates and poor convergence, whereas using data below a certain difficulty threshold substantially impairs generalization to IMO-level tasks. In addition, our case studies show that the model can devise novel auxiliary constructions compared to human solutions, exhibiting creativity in geometric reasoning.

## 2 METHOD

### 2.1 GEOMETRY PROOF LANGUAGE AND ENVIRONMENT

In previous work such as AlphaGeometry, geometric structures are defined point by point through domain-specific language (DSL). Once the construction is complete, a deductive database arithmetic reasoning (DDAR) system is employed to exhaustively search for theorems, deriving all conclusions reachable from the known facts. In this work, we build InternGeometry-DDAR, an interactive

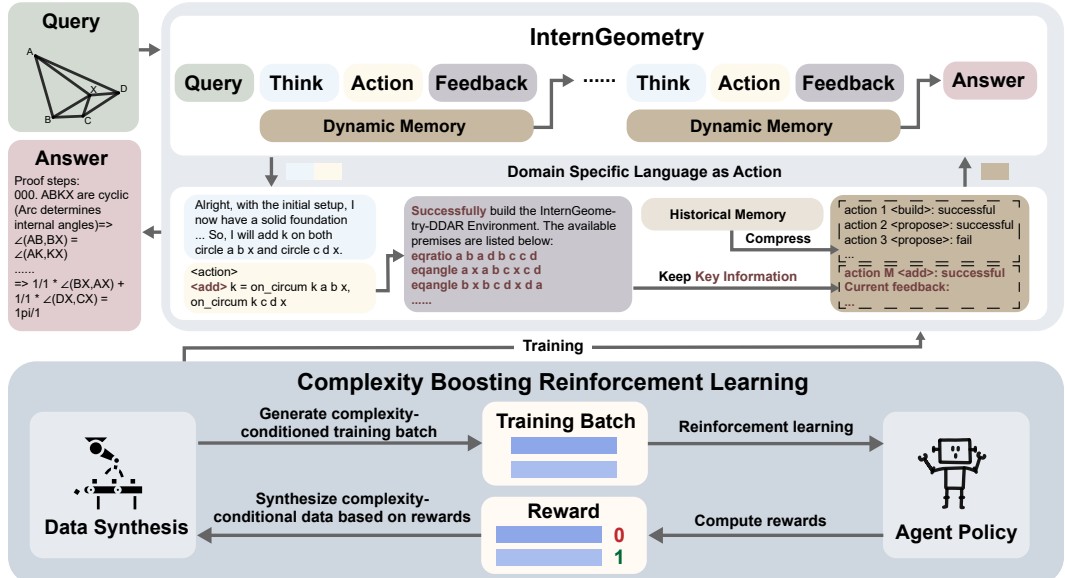

Figure 2: **An overview of InternGeometry and Complexity-Boosting Reinforcement Learning (CBRL).** (a) InternGeometry performs natural-language reasoning (Think), outputs a structured action in a domain-specific language (Action), and receives execution results (Feedback) in each turn. A dynamic memory module $\mathfrak{W}$ compresses the multi-turn interaction history to preserve essential actions and outcomes. (b) CBRL optimizes the agent policy by generating synthetic training data with controllable difficulty, assigning binary rewards to effective steps and successful outcomes, and optimizing policy through iterative reinforcement learning.

geometric proof engine based on the open-source DDAR system Newclid (Sicca et al., 2024). To support more complex geometric structures, we introduce several advanced definition strategies, such as globally optimizing point placements to satisfy constraints. During interaction, an agent not only employs the DSL for both problem specification and auxiliary point construction, but also proposes sub-proof goals that will be subsequently verified by the engine. As an interactive proof engine, InternGeometry-DDAR maintains state across steps, including the geometric configuration, constructed auxiliary points, and all proven preliminaries and propositions. Further details are provided in the Appendix B.

## 2.2 GEOMETRIC PROOF AGENT

The agent is allowed to perform natural-language reasoning at each step and then mark its final action output using an action-separating token. Let the agent be denoted by $\mathbb{G}$ and the interactive proof engine by $\mathfrak{E}$. Given a geometric problem $X$, at step $t$ the agent's output is

$$[P_t, A_t] = \mathbb{G}\left(X, \mathfrak{W}(H_{t-1})\right) \tag{1}$$

where $P_t$ denotes the slow chain-of-thought reasoning, $A_t$ the final formalized code, and $H_{t-1}$ the interaction history, which includes each round's thoughts, actions, and the feedback observations obtained from the environment prior to step $t$. The module $\mathfrak{W}$ is a memory manager that returns a compressed long history to improve the agent's long-horizon capacity.

The code $A_t$ is executed by the proof engine $\mathfrak{E}$, which is in state $t-1$. The execution result $O_t$ is appended together with the corresponding thoughts and actions to the interaction history as feedback to the agent, guiding its next step of reasoning and action.

$$O_t, \mathfrak{E}_t = \mathfrak{E}_{t-1}(A_t)$$
$$H_t = H_{t-1} + [P_t, A_t, O_t] \tag{2}$$

At each reasoning step, the agent may summarize progress, analyze the problem, or plan future proof strategies. In its action code, the agent can choose specific operations—such as constructing geometric objects, adding auxiliary constructions, or verifying whether a proposition holds.

When all targets in the problem have been proven, the geometric reasoning tool determines that the problem is fully solved, aggregates the entire proof process, and produces a complete proof of the problem, thereby concluding the reasoning session.

We posit that long-horizon capability is key to addressing the weak-heuristic challenge of auxiliary construction in geometric proofs. To this end, we introduce a dynamic memory management strategy $\mathfrak{W}$ for the agent, as well as a prior-guided rejection sampling method for $\mathbb{G}$.

To shorten the long interaction history $H$, which can span hundreds of turns, $\mathfrak{W}$ summarizes earlier exchanges, including their thoughts and detailed environment feedback, while retaining only core action outputs and key environment feedback to improve the agent's context efficiency, as illustrated in Figure 2. Through $\mathfrak{W}$, the agent obtains a concise overview of its action history and their outcomes—e.g., whether an auxiliary construction was successfully added and whether a proposed proposition holds. The last turn feedback remains unchanged, informing the agent of the currently known propositions.

Another major challenge for long-horizon agents is action collapse, where the model falls into poor patterns, such as producing highly repeated outputs or outputs similar to previous rounds (Sinha et al., 2025). To address this challenge, we use an intuitive rejection sampling method during agent inference to avoid such patterns. Denote naive LLM inference as $[\hat{P}_t, \hat{A}_t] = G\left(X, \mathfrak{W}(H_{t-1})\right)$. Then, for the sampled value $[\hat{P}_t, \hat{A}_t]$,

$$\text{If PassCheck}([\hat{P}_t, \hat{A}_t]): \quad \mathbb{G}\left(X, \mathfrak{W}(H_{t-1})\right) = [\hat{P}_t, \hat{A}_t]$$
$$\text{Else: reject the value of } [\hat{P}_t, \hat{A}_t] \text{ and return to the sampling step} \tag{3}$$

Here, PassCheck is a rule-based multi-condition checking policy that enforces no repeated actions relative to the history, no excessively long thinking without a stop, no turn without a valid action or with formatting issues, and no use of the same action type for too many consecutive rounds.

## 2.3 COMPLEXITY BOOSTING REINFORCEMENT LEARNING

Before reinforcement learning begins, there is a cold-start phase in which we perform supervised fine-tuning on a small amount of synthetic data to help the agent quickly adapt to the task paradigm.

Denote the training dataset as $\mathcal{D} = \left(X^i, h^i, y^i\right)_{i=1}^N$, where $X$ denotes the input, $h = \mathfrak{W}(H)$ denotes the compressed history (aligned with the agent's input), and $y = [P, A]$ denotes the output, including the thinking content $P$ and the action content $A$. Let the agent model $G$ have parameters $\theta$. The supervised fine-tuning objective is:

$$L_{st} = \frac{1}{N} \sum_{i=1}^N \left[ -\sum_{t=1}^T \log G_\theta\left(y_t^i \mid x^i, h_t^i\right) \right] \tag{4}$$

Based on the supervised fine-tuning result $\theta_{st}$, the agent exhibits the basic behavior patterns expected in geometric reasoning tasks, such as slow thinking and proactively invoking tools.

The subsequent CBRL phase is an iterative interaction–training loop. In each iteration, the agent first attempts a proof on the training task and then performs online learning using reward signals from its trajectories. Following GRPO (Shao et al., 2024), given task $X$, the training gradient is:

$$\nabla J_{rl}(X, \theta) = \mathbb{E}_{y, h \sim G_\theta(\cdot \mid X)}$$
$$\sum_{t=1}^T \min\left( \frac{G_\theta\left(y_t \mid X, h_t\right)}{G_{\theta_{\text{old}}}\left(y_t \mid X, h_t\right)}, \text{clip}\left(\frac{G_\theta\left(y_t \mid X, h_t\right)}{G_{\theta_{\text{old}}}\left(y_t \mid X, h_t\right)}, 1 - \epsilon, 1 + \epsilon\right) \right) A(X, y_t) \nabla G_\theta\left(y_t \mid X, h_t\right) \tag{5}$$
$$- \beta \nabla \mathbb{D}_{KL}\left(G_\theta \| G_{\text{ref}}\right)$$

where

$$A^i(X, y_t) = \frac{r_t^i - \text{mean}\left(\left\{r_1^1, r_2^1, \cdots, r_T^1, r_1^2, \cdots, r_T^K\right\}\right)}{\text{std}\left(\left\{r_1^1, r_2^1, \cdots, r_T^1, r_1^2, \cdots, r_T^K\right\}\right)}, \tag{6}$$

represents the advantage at step $t$ of the $i$-th trajectory within a batch of $K$ samples. It measures the quality improvement at step $t$ of the $i$-th generated trajectory relative to the average policy. Here, $r_t^i$ denotes the reward at step $t$ of the $i$-th trajectory. $\epsilon$ is a hyperparameter that constrains the policy

ratio. $G_{\theta_{\text{old}}}$ is the policy model from the previous iteration. $G_{\text{ref}}$ is the initial model. $\mathbb{D}_{KL}$ is the KL divergence, used as a regularizer to constrain optimization of the agent model.

Here, the reward is a binary value, computed as the conjunction of the outcome reward and the step effectiveness reward:

$$r = r^o \wedge r^s \tag{7}$$

The outcome reward $r^o$ is 1 if the proof is complete; otherwise, it is 0. The step effectiveness reward is defined by whether the step's action succeeds. For proposition-proposing steps, $r^s$ is 1 if the proposed proposition is successfully proven by the engine. For auxiliary-construction steps, $r^s$ is 1 if the construction is successfully added and used in the final proof of the overall question; otherwise, it is 0.

Note that the reward in our work is deliberately simple and can be computed by rules automatically. It rewards effective steps in trajectories that succeed while penalizing all steps in failed trajectories and ineffective steps in successful trajectories.

Next, we focus on the curriculum algorithm for CBRL in geometric tasks. One major advantage of our data synthesis pipeline is the fully controllable difficulty of the problems. As illustrated by AlphaGeometry (Trinh et al., 2024), the difficulty of IMO geometry problems for humans is positively correlated with the number of DDAR proof steps. Therefore, we choose the DDAR proof step count as the measure of task complexity, denoted as $\kappa$. Denote the data synthesis pipeline as $\mathfrak{X}$. We implement CBRL as follows:

$$\theta^* = \arg\max_{\theta} \mathbb{E}_{X \sim \mathfrak{X}(\kappa)} J_{rl}(X, \theta) \tag{8}$$

$$\kappa^* = \arg\max_{\kappa} \mathbb{E}_{X \sim \mathfrak{X}(\kappa)} \mathbb{E}_{y \sim G_{\theta}} |A(X, y)| \tag{9}$$

where the goal of $\kappa$ optimization is to maximize the average absolute advantage during learning, and $A(X, y)$ is the advantage of outcome reward. We then present properties of CBRL. As shown in Wang et al. (2025b) and Chen et al. (2025b), maximum absolute advantage has the following properties:

**Theorem 1.** *Given model parameter $\theta$, the $\kappa^*$ obtained from Equation 9 approximately optimally accelerates the learning progress.*

**Theorem 2.** *For binary rewards, the maximum average absolute advantage is 0.5, which indicates that the task is of moderate difficulty for the model—neither too difficult nor too trivial.*

In practice, in each CBRL round, we sample data conditioned on complexity $\kappa$, perform RL training to the agent, and finally update $\kappa$ according to learning rate $\alpha$. See Appendix D for details.

## 2.4 DATA SYNTHESIS PIPELINE

The proposed data pipeline targets to synthesize geometry problems with adaptable levels of difficulty for InternGeometry. Specifically, it comprises two stages: cold start and expert-level problem synthesis for CBRL.

First, due to the scarcity of data in DSL form, we fine-tuned InternThinker-32B (Lyu et al., 2025) as InternGeometry-Formalizer through expert iteration (Anthony et al., 2017) and then exploit large-scale natural language problem data from diverse sources. This process produced a total of 7K formal problem and solution trajectory pairs, which provide a cold start for InternGeometry.

However, these paired data are constrained by the imbalanced difficulty distribution with relatively few problems at the expert level. To endow LLM with expert-level problem-solving abilities, we further synthesize problems dynamically during reinforcement learning. We first add auxiliary constructions into randomly constructed problems with statistical prior of given complexity, then leveraging the InternGeometry-DDAR to filter valid constructions and goals to form new problems. Finally, a total of 6K problems are constructed by difficulty based on proof steps during CBRL. See Appendix D for details.

Table 1: Comparison of overall performance on IMO 50 between InternGeometry and SOTA geometry expert models.

| Model | Model Type | Training Data | Sampling Setting | IMO 50 Pass@K |
|---|---|---|---|---|
| AlphaGeometry 2 | Expert Model | 300M | Ensemble of search trees | 42/50 |
| SeedGeometry | Expert Model | 230M | N/A | 43/50 |
| InternGeometry | LLM Agent | 13K | Pass@256 | 44/50 |

Table 2: A problem-by-problem comparison on IMO 50 between InternGeometry and SOTA geometry expert models.

| Year | ID | Split | AG2 | SG | IG | Year | ID | Split | AG2 | SG | IG | Year | ID | Split | AG2 | SG | IG |
|---|---|---|---|---|---|---|---|---|---|---|---|---|---|---|---|---|---|
| 2000 | P1 | | ✓ | ✓ | ✓ | 2007 | P2 | | ✓ | ✓ | ✓ | 2015 | P3 | | ✓ | ✓ | ✓ |
| 2000 | P6 | | ✓ | ✓ | ✓ | 2007 | P4 | | ✓ | ✓ | ✓ | 2015 | P4 | | ✓ | ✓ | ✓ |
| 2001 | P1 | | ✗ | ✗ | ✗ | 2008 | P1 | a | ✓ | ✓ | ✓ | 2016 | P1 | | ✓ | ✓ | ✓ |
| 2001 | P5 | | ✓ | ✗ | ✓ | 2008 | P1 | b | ✓ | ✓ | ✓ | 2017 | P4 | | ✓ | ✓ | ✓ |
| 2002 | P2 | a | ✓ | ✓ | ✓ | 2008 | P6 | | ✓ | ✓ | ✓ | 2018 | P1 | | ✓ | ✓ | ✓ |
| 2002 | P2 | b | ✓ | ✓ | ✓ | 2009 | P2 | | ✓ | ✓ | ✓ | 2018 | P6 | | ✗ | ✓ | ✓ |
| 2002 | P6 | | ✗ | ✗ | ✗ | 2009 | P4 | a | ✓ | ✓ | ✓ | 2019 | P2 | | ✓ | ✓ | ✓ |
| 2003 | P3 | | ✗ | ✗ | ✗ | 2009 | P4 | b | ✓ | ✗ | ✓ | 2019 | P6 | | ✓ | ✓ | ✓ |
| 2003 | P4 | a | ✓ | ✓ | ✓ | 2010 | P2 | | ✓ | ✓ | ✓ | 2020 | P1 | | ✓ | ✓ | ✓ |
| 2003 | P4 | b | ✓ | ✓ | ✓ | 2010 | P4 | | ✓ | ✓ | ✓ | 2020 | P6 | | ✗ | ✗ | ✗ |
| 2004 | P1 | | ✓ | ✓ | ✓ | 2011 | P6 | | ✓ | ✓ | ✓ | 2021 | P3 | | ✓ | ✓ | ✓ |
| 2004 | P5 | a | ✓ | ✓ | ✓ | 2012 | P1 | | ✓ | ✓ | ✓ | 2021 | P4 | | ✓ | ✓ | ✓ |
| 2004 | P5 | b | ✓ | ✓ | ✓ | 2012 | P5 | | ✓ | ✓ | ✓ | 2022 | P4 | | ✓ | ✓ | ✓ |
| 2005 | P1 | | ✓ | ✓ | ✓ | 2013 | P3 | | ✓ | ✓ | ✓ | 2023 | P2 | | ✓ | ✓ | ✓ |
| 2005 | P5 | | ✓ | ✓ | ✓ | 2013 | P4 | | ✓ | ✓ | ✓ | 2023 | P6 | | ✗ | ✓ | ✓ |
| 2006 | P1 | | ✗ | ✓ | ✗ | 2014 | P3 | | ✓ | ✓ | ✓ | 2024 | P4 | | ✓ | ✓ | ✓ |
| 2006 | P6 | | ✗ | ✗ | ✗ | 2014 | P4 | | ✓ | ✓ | ✓ | 2025 | P2 | | N/A | ✓ | ✓ |

# 3 EXPERIMENT

## 3.1 EXPERIMENT SETUP

**Implementation.** We use InternThinker-32B (Lyu et al., 2025) as the backbone model for our method. For the agent model, we set the maximum number of steps to 200 by default, with inference hyperparameters of temperature 0.9 and top-p 0.9. During the test, the pass@K is set to 256.

**Dataset and Baselines.** We use IMO 50 (Chervonyi et al., 2025) as the test set, which includes all geometry problems from IMO 2000 to IMO 2024. We additionally evaluate InternGeometry on the geometry problem from IMO 2025, reported separately in Table 2. We use AlphaGeometry 2 (Chervonyi et al., 2025) and SeedGeometry (Chervonyi et al., 2025) as our baselines, both of which are state-of-the-art geometry proving methods based on expert models. The performance of these baselines is taken directly from the results reported in their respective papers.

## 3.2 OVERALL RESULTS

We compare the performance of InternGeometry with baselines in Table 1. InternGeometry solved 44 problems in IMO 50, surpassing AlphaGeometry 2 and SeedGeometry. The "split" in the table refers to subproblems of questions that contain multiple subquestions. Notably, InternGeometry used only 13K data points—just 0.004% of AlphaGeometry 2 and 0.006% of SeedGeometry. Furthermore, its test-time scaling budget was also far lower than AlphaGeometry 2, which use ensembles of beam search, and the reported optimal single beam tree configuration is beam size 128, branching number (samples) 32, and beam depth 4. See Appendix G for more discussion. These comparisons clearly demonstrate the potential of LLM-based agent approaches on expert-level tasks.

We list the individual results on IMO 50 in Table 2. We additionally include the geometry problem of IMO 2025 in the table. InternGeometry solved 45 out of 51 problems, covering all problems solved by AlphaGeometry 2, and additionally solving 2018 P6 and 2023 P6. Compared to SeedGeometry, it additionally solved 2001 P5 and 2009 P4b, but missed 2006 P1. Notably, the remaining unsolved problems largely involve computations that go beyond the scope of pure geometric proof, and thus fall outside the current expressive range of geometric DDAR systems. See Appendix F for cases.

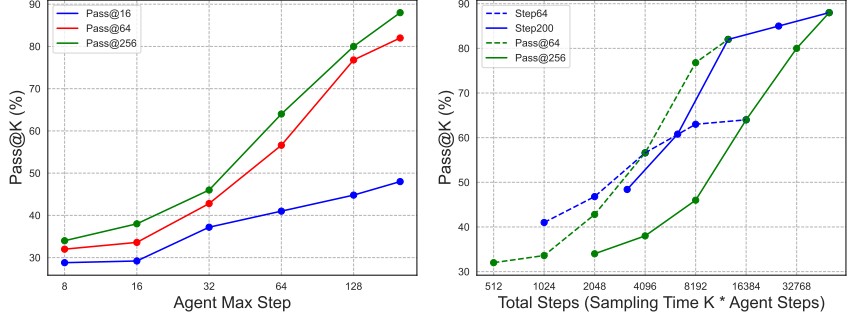

Figure 3: Left: **The effect of long-horizon interaction on the proof.** As the interaction steps increase, the proving success rate improves significantly, which holds for different sampling times. As sampling times increase, Pass@K also rises, indicating the test-time scalability of InternGeometry. Right: **Extending the agent's trajectory length is more effective than repeated sampling for scaling.** The total inference budget is defined as the sampling number K multiplied by the agent's steps. When the maximum length is capped (the blue lines), performance improves with inference budget at a slower rate for shorter trajectories. On the other hand, when the sampling size is fixed (the green lines), increasing the budget by lengthening the trajectory yields efficient scaling.

Table 3: Ablation study on long-horizon agents in InternGeometry.

| Propositions | Slow Thinking | Context Compression | Reject Sampling | IMO 50 Pass@256 |
|:---:|:---:|:---:|:---:|:---:|
| ✓ | ✓ | ✓ | ✓ | 44/50 |
| ✗ | ✓ | ✓ | ✓ | 35/50 |
| ✗ | ✗ | ✓ | ✓ | 23/50 |
| ✓ | ✓ | ✗ | ✓ | 20/50 |
| ✓ | ✓ | ✓ | ✗ | 38/50 |

### 3.3 ANALYSIS FOR LONG HORIZON AGENT

To analyze the effect of long-horizon interaction to the proof, we compare the pass@K on IMO 50 under different max step setting, and the result is in Figure 3 (left). It is evident that as interaction steps increase, the proving success rate improves significantly, which holds for different sampling times. Shorter interactions significantly limit the success rate of agent proofs. As the interaction trajectory grows, the agent can continually explore to develop heuristics about the problem, enabling it to better leverage its reasoning ability for generalization. Additionally, the figure illustrates the trend of test-time scaling. As shown, as sampling times increase, Pass@K also rises, indicating the test-time scalability of InternGeometry.

We emphasize that extending an agent's trajectory length scales performance more effectively than repeated sampling. As shown in Figure 3 (right), with total inference budget defined as $K$ times the number of steps, performance grows slowly when the step cap is 64 but improves much faster when it is 200. For fixed $K$, increasing trajectory length consistently yields higher efficiency, supporting our hypothesis that long-horizon interactions enhance heuristics more effectively in geometric proofs.

An ablation on IMO 50 (Table 3) further confirms this. InternGeometry solves fewer problems when any long-horizon component is removed. Slow thinking and context compression increase solved problems from 20 and 23 to 44, underscoring their key roles in enabling IMO-level reasoning.

### 3.4 ANALYSIS ON COMPLEXITY BOOSTING REINFORCEMENT LEARNING

In this section, we analyze the effectiveness of the CBRL method. We first present the distribution of the synthetic data obtained during the CBRL process in Figure 4 (left). As noted earlier, we use the length of a problem's proof steps as an indicator of its difficulty. Accordingly, we compile statistics on the distribution of proof lengths in the synthetic data generated during model training, as shown in the figure. The figure shows that the difficulty distribution of the synthetic data exhibits a fairly uniform improving trend, indicating that our agent training provides a well-structured curriculum from simple to difficult tasks, which helps the agent master complex combinations of proof skills.

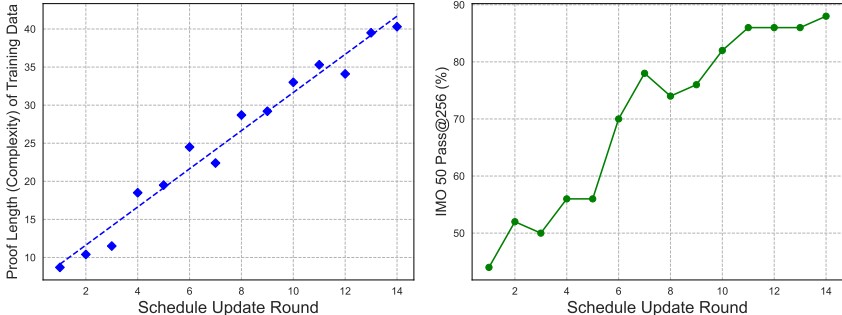

Figure 4: Left: **The distribution of proof lengths in the synthetic data generated during model training, indicating task complexity.** The figure shows that the difficulty distribution of the synthetic data exhibits a fairly uniform improving trend, providing a well-structured curriculum from simple to difficult tasks. Right: **Agent's generalization performance on IMO 50 during training.** The agent's overall performance on IMO 50 shows a steady upward trend. Notably, there is a significant performance jump in the sixth training round.

Table 4: Ablation study on CBRL in InternGeometry.

| Training Setting | IMO 50 Pass@256 |
| --- | --- |
| With CBRL | 44/50 |
| SFT Cold Start | 22/50 |
| Easy Data Only | 29/50 |
| Challenging Data Only | 24/50 |
| Same Data without Schedule | 38/50 |

We further demonstrate the agent's generalization performance on IMO 50 during training in Figure 4 (right). As the training data is continually updated, the agent's overall performance on IMO 50 shows a steady upward trend. Notably, there is a significant performance jump in the sixth training round, indicating breakthrough progress as task complexity is progressively increased in the reinforcement learning process.

We then conduct ablation studies on the key components of CBRL. The result is in Table 4. Notably, after SFT cold start, the model's baseline performance on IMO 50 is 22/50. We first analyze the impact of data difficulty on reinforcement learning, revealing the importance of allowing the learning algorithm to autonomously control the difficulty of synthesized data. To strictly control variables, we run experiments with the same data scale and training steps as in InternGeometry. Specifically, we modify the data distribution in the RL phase to: using only low-difficulty data, and using only high-difficulty data. The corresponding results are shown in the rows Easy Data Only and Challenging Data Only in the table. The results indicate that using only low-difficulty data limits the agent's generalization ability to IMO-difficulty problems; on the other hand, using only high-difficulty data also leads to suboptimal training outcomes. The latter occurs because learning becomes slow and fails to converge within the same training budget: the agent remains in an early trial-and-error stage with extremely sparse learning signals. This highlights the efficiency of CBRL.

Finally, we ablate the dynamic difficulty curriculum by uniformly sampling from all synthesized data throughout the InternGeometry training phase. The result (Same Data without Schedule) shows that removing the curriculum again degrades performance. Without progressive difficulty, the agent struggles with sparse or absent rewards on sampled hard problems, preventing it from learning effective strategies under the same training budget. This confirms that CBRL significantly improves data efficiency and training effectiveness in reinforcement learning.

## 3.5 Case Study

During our manual checking, we find that InternGeometry shows remarkable creativity on certain problems. As shown in Figure 1, while most human solvers relied on inversion, trigonometry or complex numbers, the agent solves this problem via an elegant geometric construction using classical

angle chasing and basic theorems. Specifically, InternGeometry first places a point $T$ on segment $AC$ such that $\angle BDA = \angle TDC$, and defines point $K$ as the intersection of two circles. These two points form an isogonal conjugate pair in quadrilateral $ABCD$, revealing that the agent can discover this implicit structure through exploration. To further exploit the isogonal property, it then constructs the symmetric points of $T$ with respect to each side of the quadrilateral, showing both an understanding of rotational symmetry and an ability to generalize the use of auxiliary points for handling isogonal conjugates from triangles to quadrilaterals. Overall, this case study highlights how InternGeometry generate creative constructions that differ fundamentally from human solutions.

## 4 RELATED WORK

**Reinforcement learning agents in the field of mathematics** Currently, large language model agents have achieved remarkable performance in tasks such as Code, Search, GUI, and also Mathematics through reinforcement learning (RL). In mathematics, RL agents based on informal proofs solve problems using general-purpose tools like Python compilers. Examples include OR (Open Reasoning) approaches Singh et al. (2025); Li et al. (2025b); Mai et al. (2025); Zuo et al. (2025); Prabhudesai et al. (2025); Shen et al. (2025); Shang et al. (2025) and PR (Proof Reasoning) approaches Li et al. (2025a); Simonds & Yoshiyama (2025); Goldie et al. (2025); Hao et al. (2025). Alternatively, agents built on Interactive Theorem Provers (ITPs) specialized for mathematics can handle more complex problems. Works like Xin et al. (2024); Ren et al. (2025); Zhang et al. (2025a); Wang et al. (2025a) achieve strong results on benchmarks such as miniF2F (Zheng et al., 2021) and ProofNet (Azerbayev et al., 2023). However, few agents address geometry problems. Our work targets this gap, developing an interactive geometric prover and showing the potential of data synthesis and difficulty scaling.

**Rurriculum learning for agents** While several research (Wang et al., 2025b; Zhang et al., 2025b; Parashar et al., 2025) study curriculum reinforcement learning, curriculum agents learning remains limited, and most approaches rely on highly structured task types. For example, Voyager (Wang et al., 2023) uses manually designed curricula in MineDojo (Fan et al., 2022) to teach agents complex skills. WebRL (Qi et al., 2024) iteratively generates increasingly complex task instructions and employs a reward model for automatic success evaluation. In scaling RL for agents and task synthesis, Envgen (Zala et al., 2024) trains large language models to generate formal parameters for Crafter (Hafner, 2021) and Heist (Cobbe et al., 2020), enabling dynamic environment and task generation. Unlike these approaches, our method fully automates large-scale synthesis at specified difficulty levels, allowing unrestricted and unbiased curriculum adjustment, especially at higher complexity.

**Automatic geometry theorem proving** Current AI-based approaches to automated geometric proofs remain largely expert-model driven. State-of-the-art systems like AlphaGeometry (Chervonyi et al., 2025; Trinh & Luong, 2024) and SeedGeometry (Chen et al., 2025a) typically decompose a problem into two tasks: (1) auxiliary construction prediction, where the model proposes additional geometric elements (e.g., lines, points); and (2) formal reasoning, where a search algorithm assembles a complete proof using geometric theorems (e.g., angle bisector, triangle similarity) and logical inference rules. Following this paradigm, most methods train specialized models on large datasets to predict constructions and then combine them with a formal search engine for proof generation. Recent work has also explored using large language models for geometric reasoning, but these efforts mainly target elementary-level problems and computational task formats.

## 5 CONCLUSION

LLM-based agents can solve tough math problems, even IMO-level with formal provers, but geometry remains dominated by expert systems like AlphaGeometry2 that depend on massive synthetic data and search.We introduce InternGeometry, a medalist-level LLM agent for geometry. It overcomes weak auxiliary-construction heuristics by iteratively proposing propositions and constructions, verifying them with a symbolic engine, and refining based on feedback. A dynamic memory enables over 200 interaction steps. To speed learning, we use CBRL, which gradually increases synthesized problem difficulty during training. Built on InternThinker-32B, InternGeometry solves 44/50 IMO geometry problems (2000–2024), surpassing the average gold medalist score (40.9), using only 13K training examples—about 0.004% of AlphaGeometry2's data. It can also generate novel auxiliary constructions unseen in human solutions.

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

## A    THE USE OF LARGE LANGUAGE MODELS (LLMS)

Beyond these technical roles, LLMs did not play a significant part in research ideation, experimental design, or manuscript writing, the authors conceived the study, designed/evaluated experiments, analyzed results, and wrote the paper. Any automated assistance, if present, was limited to non-substantive copy-editing/LaTeX linting. The authors take full responsibility for all contents, including verifying any machine-generated intermediate artifacts, and acknowledge that LLMs are not eligible for authorship. This disclosure follows the ICLR policy on LLM usage and research integrity.

## B    IMPROVEMENTS IN INTERNGEOMETRY-DDAR

InternGeometry-DDAR builds upon open-sourced symbolic engines (*i.e.*, Newclid (Sicca et al., 2024) and AlphaGeometry (Trinh & Luong, 2024)) and mainly consist of two components: a deductive database and algebraic reasoning. The former expands the current set of premises toward the proof goal based on geometry rules, while the latter performs angle, length, and ratio chasing using Gaussian elimination. We introduce three main improvements: dynamic diagram adjustment, the incorporation of syntax and rules for handling double points, and the addition of new predicates and rules.

First, open-sourced symbolic engines can only create points one by one based on existing construction definitions, with each point constrained by at most two construction definitions. However, in IMO geometry problem, it is often necessary to make global adjustments to previously constructed points so they satisfy more specific requirements (*e.g.*, a line defined as two existing points may also need to be tangent to a existing circle). Consider IMO 2003 P4a as an example: "Let $ABCD$ be a cyclic quadrilateral. Let $P$, $Q$ and $R$ be the feet of the perpendiculars from $D$ to the lines $BC$, $CA$ and $AB$, respectively. Show that $PQ = QR$ if the bisectors of angles $\angle ABC$ and $\angle ADC$ meet on segment $AC$". The condition "the bisectors of angles $\angle ABC$ and $\angle ADC$ meet on segment $AC$" cannot be enforced automatically during point-by-point construction; it is a specialized condition that holds only for certain point placements. Such hard configurations require globally adjusting previously constructed points so that multiple geometric constraints are satisfied simultaneously. To address this issue, we use gradient descent to adjust certain specific points so that they simultaneously satisfy multiple requirements.

Second, we address the issue of double points (*i.e.*, distinctly named points with identical coordinates) by proving that they represent the same geometric point, which is an important technique. Difficult geometry proofs often rely on reformulating intersections or handling degeneracies that naturally create such overlaps—a common trick used by human solvers. To this end, we introduce new syntax: prefixing a construction statement with ! allows the system to create a point even if it shares coordinates with an existing one. In addition, we update the inference module of the symbolic engine to support this extended behavior. We also introduce new predicates and rules for handling double points. Specifically, we define the predicate `idc x y` to indicate that points $X$ and $Y$ are geometrically considered the same, and we provide rules for determining this relationship.

Additionally, we add several common geometry theorems, such as the Power of a Point and Menelaus' theorem into InternGeometry-DDAR.

InternGeometry-DDAR serves both as an automated geometry problem solver and as a powerful symbolic tool for InternGeometry.

## C    INTERACTION BETWEEN INTERNGEOMETRY AND INTERNGEOMETRY-DDAR

InternGeometry has three interaction with InternGeometry-DDAR, which includs obtain initial state in symbolic engine, adding auxiliary construction and proposing proof steps. Specifically, when giving a formal geometry problem, InternGeometry first output '<build>' tag to construct this problem and retrieve the initial geometric relationships from the symbolic engine. Then, in the following turn, InternGeometry performs thinking and then automatically decides whether to add an auxiliary construction using '<add>' tag or propose a proof step using '<propose>'. InternGeometry-DDAR executes the instructions from InternGeometry and returns feedback, such as successfully proving the proposed proposition or reporting a failure when adding a new point. After each step, InternGeometry

summarizes and compresses the current proof state to support long-horizon interaction. When the final goal is proven, InternGeometry reviews the entire proof and briefly summarizes the reasoning process, extracting key steps and auxiliary constructions.

## D  DETAILS OF COMPLEXITY BOOSTING REINFORCEMENT LEARNING (CBRL)

We provide a detailed introduction to CBRL in this section, as the main paper is space-constrained. As outlined in Subsection 2.3, CBRL adapts task complexity to maximize the expected absolute advantage during reinforcement learning. The core intuition is to present tasks that are neither too difficult nor too easy for the policy model—i.e., tasks that best match the model's current capability. Following AlphaGeometry (Trinh & Luong, 2024), which observed that human-perceived difficulty of IMO geometry problems (measured by average IMO scores) correlates positively with the number of proof steps taken by the DDAR solver, we quantify task complexity using DDAR proof length.

We first introduce the geometry question generation algorithm, then describe the data synthesis pipeline. Finally, we present the CBRL algorithm and explain its motivation.

The question generation procedure, Generate Data (Algorithm 1), aims to sample nontrivial geometry questions that cannot be solved by exhaustive search using the InternGeometry-DDAR engine alone. The algorithm iteratively samples a raw geometric structure, $X_{raw}$, by randomly instantiating DDAR predicates and points. It then augments this structure with auxiliary constructions to obtain $X_{add}$. Both stages are conditioned on a user-specified complexity parameter $\kappa$, for which we design distinct priors and construction patterns to improve the hit rate of valid problems. We perform exhaustive search on both $X_{raw}$ and $X_{add}$. Conclusions that (i) involve only points in $X_{raw}$, (ii) are provable in $X_{add}$, and (iii) are not provable in $X_{raw}$, are deemed nontrivial problems rooted in the raw structure. Among these candidates, we select the most complex one—primarily by proof length, while also considering priors such as predicate distributions. The algorithm repeats to continuously generate questions targeting the specified complexity prior $\kappa$. Because the actual complexity of generated items is only loosely controlled by $\kappa$ via stochastic priors, we apply additional post-sampling to better align the dataset with the desired complexity.

---

**Algorithm 1** Generate Data

**Require:** Complexity $\kappa$
1: **for** _ in range(MaxSample) **do**
2:     $X_{\text{raw}} \leftarrow \text{RANDDDARCONSTRUCTION}(\kappa)$
3:     $X_{\text{add}} \leftarrow \text{ADDAUXCONSTRUCTIONS}(X_{\text{raw}}, \kappa)$
4:     $\mathcal{P}_{\text{raw}} \leftarrow \text{EXHAUSTSEARCH}(X_{\text{raw}})$
5:     $\mathcal{P}_{\text{add}} \leftarrow \text{EXHAUSTSEARCH}(X_{\text{add}})$
6:     $\mathcal{C} \leftarrow , c \in \mathcal{P}_{\text{add}} \setminus \mathcal{P}_{\text{raw}}$ & conclusion of $c$ uses only points in $X_{\text{raw}}$,
7:     **if** $\mathcal{C} \neq \emptyset$ **then**
8:         $\text{data}^\star, \text{pl}^\star \leftarrow \text{SELECTMOSTCOMPLEX}(X_{\text{raw}}, \mathcal{C})$        ▷ primarily by proof length
9:         **yield** $\text{data}^\star, \text{pl}^\star$
10:     **end if**
11: **end for**

---

Next, Algorithm 2 presents the complete Data Synthesis Pipeline. We maintain a global cache of all generated questions. Given a target complexity $\kappa$ and a required sample count, we first check whether the cache already contains enough items within a tolerance range around $\kappa$. If not, we invoke Generate Data to enrich the cache until we can retrieve a sufficient number of questions at the desired complexity.

Finally, we introduce Complexity Boosting Reinforcement Learning (CBRL) in Algorithm 3. The idea is straightforward: in each iteration, we sample a batch of data from the Data Synthesis Pipeline, run RL on that batch, and record rewards. After processing the batch, we compute the average reward and update $\kappa$ by comparing this average to $0.5$: increase $\kappa$ if the average reward exceeds $0.5$, otherwise decrease it. Below, we justify why using $0.5$ as the target lead to optimized expected absolute advantage, as indicated by Theorem 2.

---

**Algorithm 2** Data Synthesis Pipeline $\mathfrak{X}$

---

**Require:** Complexity $\kappa$, Data Number $K$
 1: **global** $C$                          ▷ cache: list of (data, proof length)
 2: $ok, \text{dataset} \leftarrow \text{SELECTAROUNDRANGE}(C, \kappa, K)$
 3: **while** not $ok$ **do**
 4:      **for** $(\text{data}, \text{pl})$ in $\text{GENERATEDATA}(\kappa)$ **do**
 5:          $C.\text{append}((\text{data}, \text{pl}))$
 6:      **end for**
 7:      $ok, \text{dataset} \leftarrow \text{SELECTAROUNDRANGE}(C, \kappa, K)$
 8: **end while**
 9: **return** dataset                   ▷ exactly $K$ items around target complexity

---

For a binary reward with $P(r = 1) = p$ and $P(r = 0) = 1 - p$, we have $\text{mean}(r) = p$ and $\text{std}(r) = \sqrt{p(1-p)}$. The expected absolute advantage is

$$\mathbb{E}[|A_i|] = \mathbb{E}\left[\left\|\frac{r_i - p}{\sqrt{p(1-p)}}\right\|\right] = p \cdot \frac{1-p}{\sqrt{p(1-p)}} + (1-p) \cdot \frac{p}{\sqrt{p(1-p)}} = 2\sqrt{p(1-p)}. \quad (10)$$

This quantity is concave in $p$ on $[0, 1]$ and is maximized at $p = 0.5$. Therefore, we can maximize the expected absolute advantage by steering the average reward toward $0.5$ via $\kappa$-adjustment.

---

**Algorithm 3** Complexity Boosting Reinforcement Learning (CBRL)

---

**Require:** Initial complexity $\kappa$, Initial policy parameter $\theta$
 1: **for** $t$ in range(MaxIter) **do**
 2:      batchdata $\leftarrow \mathfrak{X}(\kappa, \text{DataNumEachIter})$
 3:      batchrewards $\leftarrow [, ]$
 4:      **for** minibatch in $\text{SPLITMINIBATCH}(\text{batchdata})$ **do**
 5:          rewards, $\theta \leftarrow \text{EVALUATEANDUPDATE}(\theta, \text{minibatch})$
 6:          batchrewards.extend(rewards)
 7:      **end for**
 8:      **if** $\text{AVERAGE}(\text{batchrewards}) > 0.5$ **then**
 9:          $\kappa \leftarrow \kappa + \alpha$
10:      **else**
11:          $\kappa \leftarrow \kappa - \alpha$
12:      **end if**
13: **end for**
14: **return** $\theta$

---

We further explain why maximizing the expected absolute advantage benefits policy learning (Wang et al., 2025b; Chen et al., 2025b), as stated in Theorem 1. The key idea is that the expected absolute advantage serves as the primary learning signal that scales the gradient. The simplified gradient of the GRPO objective is

$$\nabla_\theta \mathcal{J}(\theta) = \mathbb{E}_{x \sim \mathfrak{X}(\kappa)}\left[\mathbb{E}_{y_i \sim G_\theta(\cdot|x)}\left[A_i \cdot \nabla_\theta \log G_\theta(y_i \mid x)\right]\right] \quad (11)$$

Thus, the gradient norm is

$$\|\nabla_\theta \mathcal{J}(\theta)\| = \mathbb{E}_{x \sim \mathfrak{X}(\kappa)}\left\|\left[\mathbb{E}_{y_i \sim G_\theta(\cdot|x)}\left[A_i \cdot \nabla_\theta \log G_\theta(y_i \mid x)\right]\right]\right\| \quad (12)$$

Assuming that the gradient term $\nabla_\theta \log G_\theta(y_i \mid x)$ is bounded and approximately random in direction, the gradient norm is approximately maximized by maximizing $\mathbb{E}_{X \sim \mathfrak{X}(\kappa)}\mathbb{E}_{y \sim G_\theta}|A(X, y)|$. In other words, increasing the expected absolute advantage yields larger gradients during optimization and thus can accelerate learning.

# E   DETAILS OF TRAINING TOKEN

We report the total number of training tokens and a token-based comparison to prior work. Our model totally trains approximately $1.91 \times 10^9$ tokens. For reference, AlphaGeometry 2 reports training on up to $1 \times 10^{12}$ tokens. Framed in terms of training tokens, InternGeometry is therefore substantially more data-efficient.

# F   FAILED CASES

We illustrate failure cases of InternGeometry on IMO 50 in this section. Notably, the unsolved problems are largely beyond pure geometry, and they primarily rely on numerical or non-geometric analysis.

**2001 P1**

Let $ABC$ be an acute-angled triangle with $O$ as its circumcenter. Let $P$ on line $BC$ be the foot of the altitude from $A$. Assume that $\angle BCA \geq \angle ABC + 30°$. Prove that $\angle CAB + \angle COP < 90°$.

**2002 P6**

Let $n \geq 3$ be a positive integer. Let $C_1, C_2, \ldots, C_n$ be unit circles in the plane, with centers $O_1, O_2, \ldots, O_n$ respectively. If no line meets more than two of the circles, prove that

$$\sum_{1 \leq i < j \leq n} \frac{1}{O_i O_j} \leq \frac{(n-1)\pi}{4}.$$

**2003 P3**

Each pair of opposite sides of a convex hexagon has the property that the distance between their midpoints is $\dfrac{\sqrt{3}}{2}$ times the sum of their lengths. Prove that the hexagon is equiangular.

**2006 P1**

Let $ABC$ be a triangle with incenter $I$. A point $P$ in the interior of the triangle satisfies

$$\angle PBA + \angle PCA = \angle PBC + \angle PCB.$$

Show that $AP \geq AI$ and that equality holds if and only if $P = I$.

**2006 P6**

Assign to each side $b$ of a convex polygon $P$ the maximum area of a triangle that has $b$ as a side and is contained in $P$. Show that the sum of the areas assigned to the sides of $P$ is at least twice the area of $P$.

**2020 P6**

Consider an integer $n > 1$, and a set $S$ of $n$ points in the plane such that the distance between any two different points in $S$ is at least 1. Prove there is a line $\ell$ separating $S$ such that the distance from any point of $S$ to $\ell$ is at least $\Omega(n^{-1/3})$.
(A line $\ell$ separates a set of points $S$ if some segment joining two points in $S$ crosses $\ell$.)

## G  Discussion of Inference Cost

Because neither AlphaGeometry 2 nor SeedGeometry has released open-source code or models, we cannot perform a direct, controlled comparison. Therefore, we rely on the configurations reported in their respective papers to provide a rough estimate and comparison. Since the SeedGeometry paper does not detail its inference budget, we primarily base our equivalent estimates on AlphaGeometry 2.

According to the AlphaGeometry 2 paper, the method uses a Shared Knowledge Ensemble of Search Trees (SKEST) that integrates classical beam search with several tree-search variants (e.g., predicting multiple auxiliary points at each node) and runs multiple trees with different search configurations and models in parallel. Consequently, the total inference budget scales with the number of configurations × the number of instantiated trees × the number of models. The reported optimal single-tree configuration uses a beam size of 128, a branching number (samples) of 32, and a beam depth of 4. However, within SKEST there are configurations that incur higher budgets, such as beam size 64 with depth 10 or beam size 512 with depth 4. The AlphaGeometry 2 model size is 3.3B parameters.

We compare inference efficiency from four perspectives: the equivalent number of solutions explored during inference, the overall inference steps, the environment execution cost, and the overall computation cost.

1. Equivalent number of solutions explored. For a single beam search, the equivalent number of explored solutions can be approximated as beam size × branching number. Under the optimal single-tree configuration, this is $128 \times 32 = 4{,}096$, and it can be larger for other configurations (e.g., beam size 512). The total number further scales with the number of configurations × the number of instantiated trees × the number of models. By contrast, InternGeometry's best-of-K inference explores only K solutions (256 in our experiments), addressing its solution-efficiency.

2. Inference steps. The optimal beam-tree configuration results in $128 \times 32 \times 4 = 16{,}384$ steps, and the same number of symbolic-engine executions per tree. Other configurations, such as beam size 64 and depth 10 ($64 \times 32 \times 10 = 20{,}480$) or beam size 512 and depth 4 ($512 \times 32 \times 4 = 65{,}536$), can require more steps. InternGeometry uses a simple pass@256 parallel-inference setting with up to 200 turns of agentic interaction per pass, totaling 51,200 steps. Overall, the per-tree inference budget of AlphaGeometry 2 is on the same order as InternGeometry. However, the total inference cost for AlphaGeometry 2 scales with the number of configurations × the number of instantiated trees × the number of models, leading to a larger overall inference steps.

3. Environment execution cost. Each step in both AlphaGeometry 2 and InternGeometry is executed in an engine. Because InternGeometry's total number of steps is smaller than the full SKEST budget of AlphaGeometry 2, it yields fewer total executions. Furthermore, each AlphaGeometry 2 step requires the DDAR system to attempt solving the entire problem, whereas each InternGeometry step either adds an auxiliary construction or attempts to prove a subgoal—operations that are less expensive than attempting to solve the whole problem.

4. Overall computation cost. InternGeometry's reasoning style and larger model size indeed increase computation. Each InternGeometry step involves natural-language reasoning, resulting in more tokens and higher compute: for IMO-50, the average number of output tokens per trajectory is 89.6K. The InternGeometry model (32B) is also larger than AlphaGeometry 2 (3.3B). Due to the unknown total cost of AlphaGeometry 2, a direct comparison is difficult. However, we emphasize that the increased computation from deeper reasoning and larger models should not be viewed as a drawback. Instead, it represents a feasible new scaling dimension—alongside training data size and the number of searched solutions—one that aligns more naturally with LLM-based approaches than with expert-model diagram.

