# OpenReview forum: "Achieving Olympia-Level Geometry Large Language Model Agent via Complexity Boosting Reinforcement Learning"
_ICLR.cc/2026/Conference — ICLR 2026 Poster_

### Official Review · Reviewer_5LSi · 2025-10-29

**Soundness:** 3
**Presentation:** 3
**Contribution:** 3
**Rating:** 6
**Confidence:** 4

**Summary:**

1. The paper introduces InternGeometry, an LLM-based agent that achieves medalist-level performance on International Mathematical Olympiad (IMO) geometry problems.
2. Unlike previous systems such as AlphaGeometry2, which rely heavily on large-scale data synthesis and heuristic search, InternGeometry integrates long-horizon reasoning, symbolic feedback, and a Complexity-Boosted Reinforcement Learning (CBRL) framework.
3. This work performs efficient training using only 13K examples (0.004% of AlphaGeometry2’s data).

**Strengths:**

1. This work introduces a long-horizon agentic reasoning paradigm for geometry, moving beyond static and propose a model toward interactive proof reasoning.
2. The complexity curriculum addresses sparse-reward issues and ensures stable reinforcement learning progression.
3. This work achieves state-of-the-art performance on IMO geometry problems with significantly less training data.

**Weaknesses:**

1. Does Pass@256 with 200 steps implies ~51K LLM–engine interactions per problem? Does this raise scalability and efficiency concerns.
2. The paper does not report inference time per problem or single-shot success rate (Pass@1); it will make the paper better if we have this presented so we can assess the model’s true reasoning efficiency.
3. Would be nice if authors can add a baseline between the pretrained InternThinker-32B and the CBRL-trained InternGeometry, this will make it clear on how much improvement gain from CBRL does the proposed method have.
4. Although case studies are insightful, a deeper analysis of failure cases (the unsolved problems in Table 2) or reasoning trajectories would provide better understanding. Authors can add this qualitative analysis in the draft, which will make the readability better.
5. Will be great if the paper can mention inference/training compute requirements to reproduce the results. This will help in reproducibility.
6. To justify the InterGeometry's generalization capabilities, authors may want to include an analysis of the cross-domain performance on mathematical and scientific reasoning tasks, which can improve the usefulness of the paper.

**Questions:**

1. How crucial are the additions of "dynamic diagram adjustment" and "double points" handling for solving the IMO-level problems? Will be good if authors can add this to improve the quality of the paper.
2. As mentioned in Equation (9) add a brief explanation of how maximizing "absolute advantage" creates a curriculum of moderate difficulty and how the search for ‘k’ is done in each round.
3. The paper builds upon InternThinker-32B as the foundation for InternGeometry model. However, the current version lacks sufficient details about InternThinker-32B’s architecture, training setup, and reasoning capabilities. Hope authors can add this to the draft and improve the paper quality.
4. Also, please clarify what aspects of InternThinker-32B enable effective long-horizon reasoning and symbolic interaction in this work.

---

> ### Author Response · Authors · 2025-11-23
> **1. Does Pass@256 with 200 steps implies ~51K LLM–engine interactions per problem? Does this raise scalability and efficiency concerns.**
>
> We thank Reviewer 5LSi for your comprehensive feedback. We are happy to address the reviewer’s concerns and will incorporate all feedback to improve the paper.
>
> Thank you for this question. We compare the interaction step number with AlphaGeometry 2. According to the AlphaGeometry 2 paper, the method uses a Shared Knowledge Ensemble of Search Trees (SKEST) that integrates classical beam search with several tree-search variants (e.g., predicting multiple auxiliary points at each node) and runs multiple trees with different search configurations and models in parallel. Consequently, the total inference budget scales with the number of configurations × the number of instantiated trees × the number of models. The reported optimal single-tree configuration uses a beam size of 128, a branching number (samples) of 32, and a beam depth of 4. However, within SKEST there are configurations that incur higher budgets, such as beam size 64 with depth 10 or beam size 512 with depth 4.
>
> The optimal beam-tree configuration results in 128 × 32 × 4 = 16,384 steps. Other configurations, such as beam size 64 and depth 10 (64 × 32 × 10 = 20,480) or beam size 512 and depth 4 (512 × 32 × 4 = 65,536), can require more steps. InternGeometry uses a simple pass@256 parallel-inference setting with up to 200 turns of agentic interaction per pass, totaling 51,200 steps. Overall, the per-tree inference budget of AlphaGeometry 2 is on the same order as InternGeometry. However, the total inference cost for AlphaGeometry 2 scales with the number of configurations × the number of instantiated trees × the number of models, making its overall inference steps larger than InternGeometry’s.
>
> Each step in both AlphaGeometry 2 and InternGeometry is executed in an engine. Because InternGeometry’s total number of steps is smaller than the full SKEST budget of AlphaGeometry 2, it yields fewer total executions. Furthermore, each AlphaGeometry 2 step requires the DDAR system to attempt solving the entire problem, whereas each InternGeometry step either adds an auxiliary construction or attempts to prove a subgoal—operations that are less expensive than attempting to solve the whole problem.
> Therefore, the engine interaction cost of InternGeometry is lower than that of AlphaGeometry 2. In practice, this can be mitigated by running all samples in parallel to speed up processing.

---

> > ### Author Response · Authors · 2025-11-23
> > **7. How crucial are the additions of "dynamic diagram adjustment" and "double points" handling for solving the IMO-level problems? Will be good if authors can add this to improve the quality of the paper.**
> >
> > Thanks for the question. Regarding dynamic diagram adjustment, consider IMO 2003 P4a as an example:
> >
> > Let $ABCD$ be a cyclic quadrilateral. Let $P$, $Q$ and $R$ be the feet of the perpendiculars from $D$ to the lines $BC$, $CA$ and $AB$, respectively. Show that $PQ = QR$ if the bisectors of angles $\angle ABC$ and $\angle ADC$ meet on segment $AC$.
> >
> > The condition "the bisectors of angles $\angle ABC$ and $\angle ADC$ meet on segment $AC$" cannot be enforced automatically during point-by-point construction; it is a specialized condition that holds only for certain point placements. Such hard configurations require globally adjusting previously constructed points so that multiple geometric constraints are satisfied simultaneously.
> >
> > In addition, support for double points (i.e., distinct symbolic points sharing the same coordinates) is crucial because difficult geometry proofs often rely on reformulating intersections or handling degeneracies that naturally create such overlaps—a common trick used by human solvers.
> >
> > Enabling these two capabilities improves coverage on complex problems. We have revised Appendix B that introduces the DDAR system to make this clearer.

---

> ### Author Response · Authors · 2025-11-23
> **2. The paper does not report inference time per problem or single-shot success rate (Pass@1); it will make the paper better if we have this presented so we can assess the model’s true reasoning efficiency.**
>
> Thank you for the suggestion. We have provided the Pass@1 distribution of IMO 50 among different intervals, as shown in the table below. Note that Pass@1=1 corresponds to the problem that can be solved by the DDAR system directly without adding auxiliary constructions.
>
> | Pass@1 Interval | (0, 0.01) | (0.01, 0.05) | [0.05, 0.2) | [0.2, 0.5) | [0.5, 0.8) | [0.8, 1) |  1 |
> |:---------------:|:---------:|:------------:|:-----------:|:----------:|:----------:|:--------:|:--:|
> |      Count      |     3     |       4      |      14     |      3     |      0     |     4    | 16 |

---

> ### Author Response · Authors · 2025-11-23
> **3. Would be nice if authors can add a baseline between the pretrained InternThinker-32B and the CBRL-trained InternGeometry, this will make it clear on how much improvement gain from CBRL does the proposed method have.**
>
> Thank you for your suggestion. Our ablation study includes the closest baseline, i.e., the SFT-cold-start, which is SFT on cold start data but not trained with CBRL. As reported in Table 4, this model solves 22/50 problems on IMO-50, which performs worse than the full CBRL-trained InternGeometry (44/50 after applying CBRL), clearly showing the performance gain contributed by CBRL.

---

> ### Author Response · Authors · 2025-11-23
> **4. Although case studies are insightful, a deeper analysis of failure cases (the unsolved problems in Table 2) or reasoning trajectories would provide better understanding. Authors can add this qualitative analysis in the draft, which will make the readability better.**
>
> We thank the reviewer for the suggestion. In the revised paper, we have added an analysis of the failure cases in Appendix E. Notably, the unsolved problems are largely beyond pure geometry, and they primarily rely on numerical or non-geometric analysis, which makes them inherently difficult to formalize and be addressed by the current DDAR systems. Take 2002 P6 as an example:
>
> Let $n \ge 3$ be a positive integer. Let $C_1, C_2, \ldots, C_n$ be unit circles in the plane, with centers $O_1, O_2, \ldots, O_n$ respectively. If no line meets more than two of the circles, prove that
> $\sum_{1 \le i < j \le n} \frac{1}{O_iO_j} \le \frac{(n-1)\pi}{4}.$

---

> ### Author Response · Authors · 2025-11-23
> **5. Will be great if the paper can mention inference/training compute requirements to reproduce the results. This will help in reproducibility.**
>
> Thank you for your suggestion. During inference, the average number of output tokens per trajectory is 89.6K for IMO-50. We run pass@256 sampling for each question by running all the trials in parallel.
>
> We also report the total number of training tokens and a token-based comparison to prior work. Our model totally trains approximately 1.91×10^9 tokens. For reference, AlphaGeometry 2 reports training on up to 1×10^12 tokens. However, the major cost during training is in the rollout on training data, which relies on large-scale parallel inference.

---

> ### Author Response · Authors · 2025-11-23
> **6. To justify InterGeometry's generalization capabilities, authors may want to include an analysis of the cross-domain performance on mathematical and scientific reasoning tasks, which can improve the usefulness of the paper.**
>
> Thanks for the question. We additionally trained a version of the model on geometry agentic proof, together with agentic informal mathematical reasoning by multi-turn reasoning and summarization. This variant preserves the geometry agent’s capabilities and achieves strong performance on informal math tasks via multi-turn agentic reasoning. We report its agentic informal reasoning performance on mathematical benchmarks beyond IMO-AG-50, summarized in the table below. It performs strongly on non-geometry tasks as well.
>
> | Benchmarks | AIME2025  | HMMT2025  |
> |:----------:|:---------:|:---------:|
> |   Pass@1   |   86.67   |   71.04   |

---

> ### Author Response · Authors · 2025-11-23
> **8. As mentioned in Equation (9) add a brief explanation of how maximizing "absolute advantage" creates a curriculum of moderate difficulty and how the search for ‘k’ is done in each round.**
>
> Thanks for the question. We provide a detailed introduction to CBRL in Appendix D, as the main paper is space-constrained. The core intuition is to present tasks that are neither too difficult nor too easy for the policy model—i.e., tasks that best match the model’s current capability. Following AlphaGeometry, which observed that human-perceived difficulty of IMO geometry problems (measured by average IMO scores) correlates positively with the number of proof steps taken by the DDAR solver, we quantify task complexity using DDAR proof length.
>
> Appendix D first introduced the Generate Data algorithm and Data Synthesis Pipeline, which allow complexity prior-guided data generation and post filtering. Based on Data Synthesis Pipeline, it is feasible to streamingly generate training data with expected complexity during training.
>
> We then introduce CBRL in Appendix D in detail. As outlined in Subsection 2.3, CBRL adapts task complexity to maximize the expected absolute advantage during reinforcement learning. For binary rewards that P(r=1) = p, it is easy to show the goal of maximizing expected absolute advantage is achieved at p=0.5:
> $$
> \mathbb{E}\left[\left|A_i\right|\right]
> = \mathbb{E}\left[\left|\frac{r_i - p}{\sqrt{p(1-p)}}\right|\right]
> = p\cdot\frac{1-p}{\sqrt{p(1-p)}} + (1-p)\cdot\frac{p}{\sqrt{p(1-p)}}
> = 2\sqrt{p(1-p)}.
> $$
>
> This quantity is concave in p on [0,1] and is maximized at $p=0.5$. Therefore, we can maximize the expected absolute advantage by steering the average reward toward 0.5 via $\kappa$-adjustment. During each CBRL iteration, data is sampled with complexity $\kappa$. The policy model is run on the data, and we compare the average reward (i.e., p) to 0.5. We increase $\kappa$ by $\alpha$ if p>0.5, otherwise decrease $\kappa$. This goal can be simply explained as providing training tasks that are neither too easy nor too hard for the policy model, which would intuitively speed up the training.

---

> ### Author Response · Authors · 2025-11-23
> **9. The paper builds upon InternThinker-32B as the foundation for InternGeometry model. However, the current version lacks sufficient details about InternThinker-32B’s architecture, training setup, and reasoning capabilities. Hope authors can add this to the draft and improve the paper quality.**
>
> We apologize for the confusion caused by the model name. In this paper, we use InternThinker-32B, which is also referred to as OREAL-32B and is available on HuggingFace. Specifically, the model is built on the Qwen2.5-32B architecture and trained with Outcome REwArd-based Reinforcement Learning (OREAL) [1], which amplifies key reasoning steps while maintaining stability in long-chain reasoning and demonstrates strong performance on mathematical benchmarks such as AIME 2025 and OlympiadBench, suitable for our senario.
>
> [1] Chengqi Lyu et al. Exploring the Limit of Outcome Reward for Learning Mathematical Reasoning.

---

> ### Author Response · Authors · 2025-11-23
> **10. Also, please clarify what aspects of InternThinker-32B enable effective long-horizon reasoning and symbolic interaction in this work.**
>
> We choose InternThinker-32B as it is trained with outcome reward-based reinforcement learning to boost long-chain and slow reasoning capacity, and demonstrate strong performance on mathematical benchmarks such as AIME 2025 and OlympiadBench, which is suitable for our senario.
>
> In addition, we evaluate our method using other backbone models to assess how backbone capacity influences geometric reasoning and to examine the generality of our CBRL framework across different LLMs. Specifically, we report results for QWQ-32B and Intern-s1-mini (8B) on the IMO-50 benchmark under a consistent evaluation protocol (Pass@256). All of these models are available on Hugging Face.
>
> |       Backbone      | IMO-50 (Pass@256) |
> |:-------------------:|:-----------------:|
> |  InternThinker-32B  |       44/50       |
> |       QWQ-32B       |       44/50       |
> | Intern-s1-mini (8B) |       40/50       |
>
> These results yield two key observations. First, the 32B backbones substantially outperform the 8B mini backbone (44/50 vs. 40/50), indicating that larger models offer stronger geometric reasoning capabilities in our setting. Second, QWQ-32B attains the same score as InternThinker-32B, suggesting that our CBRL method is not tied to particular backbone and generalizes well across different backbones.

---

### Official Review · Reviewer_2RDT · 2025-11-03

**Soundness:** 3
**Presentation:** 3
**Contribution:** 4
**Rating:** 8
**Confidence:** 4

**Summary:**

While LLM agents and agentic workflows have shown strong performance (gold medal level) in International Mathematical Olympiad (IMO), solving geometry problems is still by far the most challenging task, and existing IMO math agents typically have poor performance on such problems. The main reason is that a large subset of geometry problems typically requires multiple trial and error to find the right auxiliaries to add to the current problem diagram to make it solvable. Solutions like AlphaGeometry exist, these solutions typically require an enormous amount of formal data to either train a custom transformer from scratch or perform continual training on the same data but on already pre-trained models. Then, the trained model on formal data would go in a loop with a deductive symbolic engine, where the model proposes constructing a new auxiliary to the problem at hand, and then the symbolic engine would run over and over again to extensively perform deductive search for reaching the proof goal, and upon exhaustion of the engine, the loop continues.

The vast amount of search done on formal deductive engines, and the enormous amount of training data make the existing solutions very inefficient. On the other hand, it is widely known that LLM agents and workflows can reach perfect performance on non-geometry problems in either formal or non-formal settings even when fine-tuned on a very limited amount of training data compared to AlphaGeometry. Thus, this work aims to address this timely question of how LLM agents can be designed to tackle geometry problems as well. They propose introducing a novel tool call for dynamic interaction with their enhanced deductive engine, supporting multiple actions such as building the formal geometric configuration, adding auxiliary constructions, and proposing a proposition to be verified. The dynamic interaction with the engine enables leveraging the already strong high-level natural language planning and reasoning of frontier LLMs for optimized interaction with the engine.

To enable this tool call, they performed cold-start training to teach their base model (InternThinker-32B) how to work with this new tool, and then performed curriculum RL to improve the reasoning with their proposed tool call, and the final solution is InternGeometry which is claimed to perform on par with, or even sometimes outperform the two main baselines: AlphaGeometry and SeedGeometry on the IMO 50 evaluation suite (all geometry problems from 2000-2024 IMO). In contrast to these baselines, InternGeometry is trained on roughly 0.005% of the training data size of other baselines (13k samples, 7k for cold-start, and 6k for RL).

**Strengths:**

1. The introduction of a new tool call for leveraging strong planning and reasoning of LLMs for dynamic interaction with the formal deductive engine is novel and an exciting idea. The paper did a good job explaining this tool call and motivating it, which makes their high-level approach clear and sound.

2. As mentioned in the summary, much trial and error is typically expected for figuring out the helpful auxiliary constructions for solving the given problem, and this fact would require the proposed agent to deal with the challenge of long-horizon reasoning that demands careful context management of past exploration after each tool call. The paper then proposes to maintain a dynamic memory containing key information from past tool call iterations (e.g., what actions were made in previous turns, what were the outcomes of those actions, plus the current action and current feedback from the deductive engine which contains all successful propositions). The introduction of this dynamic memory is novel and an interesting solution for solving the weak heuristic nature of auxiliary construction in IMO-level geometry problem solving. The ablation also confirms the long-horizon interaction is indeed necessary for obtaining the InternGeometry performance.

3. Only requiring roughly 0.005% of the key baselines' training data (AlphaGeometry and SeedGeometry) to reach comparable performance, and even outperform them on the IMO 50 dataset is an impactful contribution to the LLM math agents community, and this work can be seen as an initial step to harness general-purpose LLMs for solving complex geometry problems.

4. The paper mentions the model, data, and the deductive engine used will be open-sourced, which is of great benefit to the community, and this would make this work reproducible.

**Weaknesses:**

1. While the RL reward and RL loss are clearly defined, the handling and explanation of the curriculum algorithm lacks clarity, which makes it hard to evaluate the soundness of the proposed curriculum approach. The paper attempts to touch on the theory behind the curriculum algorithm on the surface, and both Theorem 1 and 2 statements are hard to follow and vague, and could be better explained. More importantly, the paper does not explain the CBRL algorithm with sufficient detail and particularly it is not clear how the complexity $\kappa$ is updated in each CBRL round. The paper only briefly mentions the following (line 253): "In practice, in each CBRL round, we sample data conditioned on complexity $\kappa$, perform RL training to the agent, and finally update κ according to learning rate $\alpha$."

2. Data curation for cold-start requires clarity, as the complexity of obtaining and curating such data is actually high. Regarding this, the paper only briefly addresses this around line 262, mentioning that "First, due to the scarcity of data in formal systems, we fine-tuned InternThinker-32B as InternGeometry-Formalizer through expert iteration (Anthony et al., 2017) and then exploit large-scale natural language problem data from diverse sources. This process produced a total of 7K formal problem and solution trajectory pairs, which provide a cold start for InternGeometry." I believe due to the complexity of data curation for this phase, one would want to know the details of how exactly the cold-start trajectory is generated.

**Questions:**

1. Could you provide a simplified high-level pseudocode for the working implementation of CBRL so the selection of the curriculum complexity $\kappa$ becomes clear, as well as explaining the technical challenges of implementing the CBRL algorithm? This would greatly benefit the soundness and clarity of the RL training done in this paper.

2. Please see Weakness 2, and elaborate more on how exactly the cold-start data is generated?

3. A cost comparison between deploying InternGeometry versus AlphaGeometry or SeedGeometry would be greatly valuable for this work. I understand this might or might not be feasible based on the information available from the prior work, but it would be interesting to compare the number of interaction steps between the deductive engine and the LLM with the prior work. Especially since the LLM usage for proposing propositions might greatly improve the interaction efficiency as well.

---

> ### Author Response · Authors · 2025-11-23
> **1. While the RL reward and RL loss are clearly defined, the handling and explanation of the curriculum algorithm lacks clarity, which makes it hard to evaluate the soundness of the proposed curriculum approach.**
>
> We thank Reviewer 2RDT for your positive feedback. We are pleased to answer reviewer’s concerns below and will incorporate all feedback to polish up the paper.
>
> Thanks for the question. We provide a detailed introduction to CBRL in Appendix D in the revised paper, as the main paper is space-constrained. The core intuition is to present tasks that are neither too difficult nor too easy for the policy model—i.e., tasks that best match the model’s current capability. Following AlphaGeometry, which observed that human-perceived difficulty of IMO geometry problems (measured by average IMO scores) correlates positively with the number of proof steps taken by the DDAR solver, we quantify task complexity using DDAR proof length.
>
> Appendix D first introduced the Generate Data algorithm and Data Synthesis Pipeline, which allow complexity prior-guided data generation and post filtering. Based on Data Synthesis Pipeline, it is feasible to streamingly generate training data with expected complexity during training.
>
> We then introduce CBRL in Appendix D in detail. As outlined in Subsection 2.3, CBRL adapts task complexity to maximize the expected absolute advantage during reinforcement learning. For binary rewards that P(r=1) = p, it is easy to show the goal of maximizing expected absolute advantage is achieved at p=0.5:
> $$
> \mathbb{E}\left[\left|A_i\right|\right]
> = \mathbb{E}\left[\left|\frac{r_i - p}{\sqrt{p(1-p)}}\right|\right]
> = p\cdot\frac{1-p}{\sqrt{p(1-p)}} + (1-p)\cdot\frac{p}{\sqrt{p(1-p)}}
> = 2\sqrt{p(1-p)}.
> $$
> This quantity is concave in p on [0,1] and is maximized at $p=0.5$. Therefore, we can maximize the expected absolute advantage by steering the average reward toward 0.5 via $\kappa$-adjustment. During each CBRL iteration, data is sampled with complexity $\kappa$. The policy model is run on the data, and we compare the average reward (i.e., p) to 0.5. We increase $\kappa$ by $\alpha$ if p>0.5, otherwise decrease $\kappa$. This goal can be simply explained as providing training tasks that are neither too easy nor too hard for the policy model, which would intuitively speed up the training.

---

> ### Author Response · Authors · 2025-11-23
> **2. Data curation for cold-start requires clarity**
>
> Thanks for the question. We construct the cold-start trajectories as follows. We begin by using InternGeometry-Formalizer to convert a diverse collection of geometry problems into their formal representations. We then employ the DDAR system to solve these formalized problems along with the required auxiliary constructions, which yields complete problem–proof graphs.
>
> From each proof graph, we extract the key nodes and treat them as simulated "proposing" steps that capture the essential propositions used in the proof. We also insert the corresponding auxiliary-construction steps into the trajectory before the proof steps that depend on them.
>
> After assembling these simulated formalized steps, we replay them in the interactive InternGeometry-DDAR environment to obtain the environment feedback after each step. The trajectories are then provided to the LLM step by step, prompting it to simulate the reasoning for each step. Finally, we process these trajectories using our memory management module to produce online input–output pairs that serve as cold-start data.

---

> ### Author Response · Authors · 2025-11-23
> **3. Could you provide a simplified high-level pseudocode for the working implementation of CBRL so the selection of the curriculum complexity  becomes clear, as well as explaining the technical challenges of implementing the CBRL algorithm? This would greatly benefit the soundness and clarity of the RL training done in this paper.**
>
> Thanks for this suggestion. We provide a detailed description of CBRL in Appendix D, and we describe the overall idea of CBRL in response to Question 1.
>
> Generally, the main technical challenge of applying CBRL lies in obtaining high-quality training tasks with the required level of complexity, especially for high complexity. In broad domains, this is difficult because either
> - (a) their curriculum relies on sampling tasks from a fixed data pool, which introduces intrinsic distributional limitations and biases, or
> - (b) synthesizing high‑quality and complex data for these domains is very challenging.
>
> In our geometry setting, we address this challenge with a scalable data‑synthesis method that generates diverse, controllable, and increasingly complex tasks, mitigating both the data‑pool bias and the synthesis bottleneck.

---

> ### Author Response · Authors · 2025-11-23
> **4. A cost comparison between deploying InternGeometry versus AlphaGeometry or SeedGeometry would be greatly valuable for this work. I understand this might or might not be feasible based on the information available from the prior work, but it would be interesting to compare the number of interaction steps between the deductive engine and the LLM with the prior work. Especially since the LLM usage for proposing propositions might greatly improve the interaction efficiency as well.**
>
> Thanks for the suggestion. Because neither AlphaGeometry 2 nor SeedGeometry has released open-source code or models, we cannot perform a direct, controlled comparison. Therefore, we rely on the configurations reported in their respective papers to provide a rough estimate and comparison. Since the SeedGeometry paper does not detail its inference budget, we primarily base our equivalent estimates on AlphaGeometry 2.
>
> According to the AlphaGeometry 2 paper, the method uses a Shared Knowledge Ensemble of Search Trees (SKEST) that integrates classical beam search with several tree-search variants (e.g., predicting multiple auxiliary points at each node) and runs multiple trees with different search configurations and models in parallel. Consequently, the total inference budget scales with the number of configurations × the number of instantiated trees × the number of models. The reported optimal single-tree configuration uses a beam size of 128, a branching number (samples) of 32, and a beam depth of 4. However, within SKEST there are configurations that incur higher budgets, such as beam size 64 with depth 10 or beam size 512 with depth 4. The AlphaGeometry 2 model size is 3.3B parameters.
>
> We compare inference efficiency from four perspectives: the equivalent number of solutions explored during inference, the overall inference steps, the environment execution cost, and the overall computation cost.
>
> 1. **Equivalent number of solutions explored**. For a single beam search, the equivalent number of explored solutions can be approximated as beam size × branching number. Under the optimal single-tree configuration, this is 128 × 32 = 4,096, and it can be larger for other configurations (e.g., beam size 512). The total number further scales with the number of configurations × the number of instantiated trees × the number of models. By contrast, InternGeometry’s best-of-K inference explores only K solutions—256 in our experiments—so InternGeometry is more solution-efficient.
>
> 2. **Inference steps**. The optimal beam-tree configuration results in 128 × 32 × 4 = 16,384 steps. Other configurations, such as beam size 64 and depth 10 (64 × 32 × 10 = 20,480) or beam size 512 and depth 4 (512 × 32 × 4 = 65,536), can require more steps. InternGeometry uses a simple pass@256 parallel-inference setting with up to 200 turns of agentic interaction per pass, totaling 51,200 steps. Overall, the per-tree inference budget of AlphaGeometry 2 is on the same order as InternGeometry. However, the total inference cost for AlphaGeometry 2 scales with the number of configurations × the number of instantiated trees × the number of models, making its overall inference steps larger than InternGeometry’s.
>
> 3. **Environment execution cost**. Each step in both AlphaGeometry 2 and InternGeometry is executed in an engine. Because InternGeometry’s total number of steps is smaller than the full SKEST budget of AlphaGeometry 2, it yields fewer total executions. Furthermore, each AlphaGeometry 2 step requires the DDAR system to attempt solving the entire problem, whereas each InternGeometry step either adds an auxiliary construction or attempts to prove a subgoal—operations that are less expensive than attempting to solve the whole problem.
>
> 4. **Overall computation cost**. We acknowledge that InternGeometry’s reasoning style and larger model size increase computation. Each InternGeometry step involves natural-language reasoning, resulting in more tokens and higher compute: for IMO-50, the average number of output tokens per trajectory is 89.6K. The InternGeometry model (32B) is also larger than AlphaGeometry 2 (3.3B). Due to the unknown total cost of AlphaGeometry 2, a direct comparison is difficult. However, we emphasize that the increased computation from deeper reasoning and larger models should not be viewed as a drawback. Instead, it represents a feasible new scaling dimension—alongside training data size and the number of searched solutions—one that aligns more naturally with LLM-based approaches than with expert-model paradigm.
>
> We have also added the discussion in Appendix G.
>
> Regarding the question “Does using the LLM to propose propositions greatly improve interaction efficiency?”, we would like to clarify that althouth overall interaction steps is fewer than AG2 (point 2), LLM-based proposition generation is not intended to directly decrease the number of interaction steps, because it is itself a step that must be executed in the engine. Instead, it serves as an effective interactive reasoning mechanism that improves LLM's understanding of the problem and improves solution efficiency (point 1: Equivalent number of solutions explored), as shown by the ablation study in Table 3. Therefore, the proposing steps increase the cost of each trajectory and the overall number of interaction steps on the surface, but they significantly improve success rate and solution efficiency.

---

### Official Review · Reviewer_YNAr · 2025-11-09

**Soundness:** 3
**Presentation:** 2
**Contribution:** 3
**Rating:** 6
**Confidence:** 3

**Summary:**

The paper introduces InternGeometry, an LLM agent system with a symbolic geometry solver (InternGeometry-DDAR) that solves math competition geometry problems. It improves the previous SOTA on 50 IMO geometry problems from 43/50 to 44/50, and achieves this with significantly less geometry problem training data than previous methods.

**Strengths:**

1. InternGeometry is the new SOTA on IMO 50.
2. The fact that the training set of geometry problems is much smaller than in previous approaches is impressive.
3. The analysis about scaling max steps vs. scaling # samples is insightful.

**Weaknesses:**

1. The paper would benefit from more discussion regarding inference-time costs of the compared methods. For example, listing the model sizes in Table 1 would be helpful, as well as explaining the different search parameters in AlphaGeometry2’s custom beam search. If available, information about the total # of output tokens or wallclock time etc. would also be appreciated.

2. Similarly, I think it would be nice if the paper also discussed training costs and compared them with previous methods. Currently, it mentions the size of the training set, but other information would also be helpful, such as the total # of tokens in the training set and a version of Figure 4 where the x-axis is the number of training tokens.

3. Writing quality is poor:
  - Contextualization within previous work is often missing, e.g., it is not clear what the novelty of CBRL is compared to previous work (curriculum learning for RL, GRPO).
  - There are many grammatical mistakes and unidiomatic uses of English, including in the title and abstract. I recommend using something like ChatGPT to improve the writing.
  - There are missing citations (e.g., the last two paragraphs of the introduction section have no citations)
  - Other miscellaneous issues, e.g., labeling imprecise claims as “theorems” (lines 247-251), potential typos in Eq. (5), and missing explanations (e.g., what’s InternThinker-32B? what’s “split” in Table 2?)

**Questions:**

1. What is InternThinker-32B? There is no citation, and I couldn’t find any information about it on the Internet.

2. How are Eqs. (5) and (6) different from GRPO? (other than what appears to be typos)

3. How is CBRL different from RL with a curriculum?

4. Should the top-left cell of Table 1 say “AlphaGeometry2” instead of “AlphaGeometry”?

5. The claim that InternGeometry’s test-time scaling budget is “far lower than that of AlphaGeometry2” (lines 311-312) is not clear to me. InternGeometry does pass@256 and uses a larger model (32B instead of 3.3B), so it’s not clear that it uses "far less" computational resources than AlphaGeometry2.

---

> ### Author Response · Authors · 2025-11-23
> **1. The paper would benefit from more discussion regarding inference-time costs of the compared methods. For example, listing the model sizes in Table 1 would be helpful, as well as explaining the different search parameters in AlphaGeometry2’s custom beam search. If available, information about the total # of output tokens or wallclock time etc. would also be appreciated.**
>
> We thank the Reviewer YNAr for your comprehensive feedback. We are delighted to answer reviewer's concerns below to refine our paper.
>
>
> Thanks for the suggestion. Because neither AlphaGeometry 2 nor SeedGeometry has released open-source code or models, we cannot perform a direct, controlled comparison. Therefore, we rely on the configurations reported in their respective papers to provide a rough estimate and comparison. Since the SeedGeometry paper does not detail its inference budget, we primarily base our equivalent estimates on AlphaGeometry 2.
>
> According to the AlphaGeometry 2 paper, the method uses a Shared Knowledge Ensemble of Search Trees (SKEST) that integrates classical beam search with several tree-search variants (e.g., predicting multiple auxiliary points at each node) and runs multiple trees with different search configurations and models in parallel. Consequently, the total inference budget scales with the number of configurations × the number of instantiated trees × the number of models. The reported optimal single-tree configuration uses a beam size of 128, a branching number (samples) of 32, and a beam depth of 4. However, within SKEST there are configurations that incur higher budgets, such as beam size 64 with depth 10 or beam size 512 with depth 4. The AlphaGeometry 2 model size is 3.3B parameters.
>
> We compare inference efficiency from four perspectives: the equivalent number of solutions explored during inference, the overall inference steps, the environment execution cost, and the overall computation cost.
>
> 1. **Equivalent number of solutions explored**. For a single beam search, the equivalent number of explored solutions can be approximated as beam size × branching number. Under the optimal single-tree configuration, this is 128 × 32 = 4,096, and it can be larger for other configurations (e.g., beam size 512). The total number further scales with the number of configurations × the number of instantiated trees × the number of models. By contrast, InternGeometry’s best-of-K inference explores only K solutions—256 in our experiments—so InternGeometry is more solution-efficient.
>
> 2. **Inference steps**. The optimal beam-tree configuration results in 128 × 32 × 4 = 16,384 steps. Other configurations, such as beam size 64 and depth 10 (64 × 32 × 10 = 20,480) or beam size 512 and depth 4 (512 × 32 × 4 = 65,536), can require more steps. InternGeometry uses a simple pass@256 parallel-inference setting with up to 200 turns of agentic interaction per pass, totaling 51,200 steps. Overall, the per-tree inference budget of AlphaGeometry 2 is on the same order as InternGeometry. However, the total inference cost for AlphaGeometry 2 scales with the number of configurations × the number of instantiated trees × the number of models, making its overall inference steps larger than InternGeometry’s.
>
> 3. **Environment execution cost**. Each step in both AlphaGeometry 2 and InternGeometry is executed in an engine. Because InternGeometry’s total number of steps is smaller than the full SKEST budget of AlphaGeometry 2, it yields fewer total executions. Furthermore, each AlphaGeometry 2 step requires the DDAR system to attempt solving the entire problem, whereas each InternGeometry step either adds an auxiliary construction or attempts to prove a subgoal—operations that are less expensive than attempting to solve the whole problem.
>
> 4. **Overall computation cost**. We acknowledge that InternGeometry’s reasoning style and larger model size increase computation. Each InternGeometry step involves natural-language reasoning, resulting in more tokens and higher compute: for IMO-50, the average number of output tokens per trajectory is 89.6K. The InternGeometry model (32B) is also larger than AlphaGeometry 2 (3.3B). Due to the unknown total cost of AlphaGeometry 2, a direct comparison is difficult. However, we emphasize that the increased computation from deeper reasoning and larger models should not be viewed as a drawback. Instead, it represents a feasible new scaling dimension—alongside training data size and the number of searched solutions—one that aligns more naturally with LLM-based approaches than with expert-model paradigm.
>
> We have also added the discussion in Appendix G.

---

> > ### Author Response · Authors · 2025-11-23
> > **7. Should the top-left cell of Table 1 say “AlphaGeometry2” instead of “AlphaGeometry”?**
> >
> > Thank you for pointing out this issue. We have revised this paper in the revised paper, and it should be AlphaGeometry 2.

---

> ### Author Response · Authors · 2025-11-23
> **2. Similarly, I think it would be nice if the paper also discussed training costs and compared them with previous methods. Currently, it mentions the size of the training set, but other information would also be helpful, such as the total # of tokens in the training set and a version of Figure 4 where the x-axis is the number of training tokens.**
>
> Thanks for the thoughtful suggestion. We have added the total number of training tokens and a token-based comparison to prior work.
>
> - Our model totally trains approximately 1.91×10^9 tokens.
>
> - For reference, AlphaGeometry 2 reports training on up to 1×10^12 tokens.
>
> Framed in terms of training tokens, InternGeometry is therefore substantially more data-efficient. In the revision, we report the total training tokens alongside dataset size in Appendix E.

---

> ### Author Response · Authors · 2025-11-23
> **3. The writing quality is poor: - There are many grammatical mistakes and unidiomatic uses of English, including in the title and abstract. I recommend using something like ChatGPT to improve the writing. - There are missing citations (e.g., the last two paragraphs of the introduction section have no citations) - Other miscellaneous issues, e.g., labeling imprecise claims as “theorems” (lines 247-251), potential typos in Eq. (5), and missing explanations( what’s “split” in Table 2?)**
>
> Thank you very much for the detailed and constructive feedback.
>
> - We have carefully revised the paper for grammar and style, and added appropriate citations to previously uncited paragraphs in the introduction.
>
> - Because the title cannot be changed while the manuscript is under review, we will revise it as soon as title changes are permitted.
>
> - We have corrected typographical errors in Eq 5 and clarified theorem statements in Appendix D.
>
> - “Split” in Table 2: some problems contain multiple proof targets (such as necessary and sufficient conditions) and are thus evaluated separately.
>
> - We respond to other questions individually below.
>
> We thank the reviewer again for the helpful comments.

---

> ### Author Response · Authors · 2025-11-23
> **4. What is InternThinker-32B? There is no citation, and I couldn’t find any information about it on the Internet.**
>
> We apologize for the confusion caused by the model name. In this paper, we use InternThinker-32B, which is also referred to as OREAL-32B and is available on HuggingFace. Specifically, the model is built on the Qwen2.5-32B architecture and trained with Outcome REwArd-based Reinforcement Learning (OREAL) [1], which amplifies key reasoning steps while maintaining stability in long-chain reasoning and demonstrates strong performance on mathematical benchmarks such as AIME 2025 and OlympiadBench, suitable for our senario.
>
> [1] Chengqi Lyu et al. Exploring the Limit of Outcome Reward for Learning Mathematical Reasoning.

---

> ### Author Response · Authors · 2025-11-23
> **5. How are Eqs. (5) and (6) different from GRPO? (other than what appears to be typos)**
>
> Thank you for the question. In this work, our RL objective largely follows the GRPO formulation, with adaptations for a multi-turn agent. Specifically, Eqs. (5) and (6) are modified to handle long-horizon agent–tool interactions of up to 200 turns, as reflected in Eq. (5). We process the entire trajectory into T input–output pairs using a per-turn history compression method, and compute advantages at the step level across trajectories. This differs from the standard GRPO setup, which is primarily designed for single-turn responses.
>
> To avoid confusion, we clarify that the RL objective is not the main contribution of this work, and we do not claim novelty for it in the draft. In the revised version, we add an explicit citation to GRPO before Eq. (5) to better ground the RL formulation.

---

> ### Author Response · Authors · 2025-11-23
> **6. How is CBRL different from RL with a curriculum?**
>
> Thank you for the question. We discuss curriculum reinforcement learning in Section 4 (Related Work), under “Curriculum Learning for Agents.” Compared to prior work, our CBRL method specifically leverages a scalable data-synthesis approach tailored to the geometry domain. In broad domains, curriculum RL is difficult because: (a) many curricula sample tasks from a fixed data pool, which introduces intrinsic distributional limitations and biases—especially at higher complexities; or (b) synthesizing high-quality, complex data is itself very challenging. In our geometry setting, we address these issues with a scalable data-synthesis method that generates diverse, controllable, and increasingly complex tasks, mitigating both data-pool bias and the data-synthesis bottleneck.

---

> ### Author Response · Authors · 2025-11-23
> **8. The claim that InternGeometry’s test-time scaling budget is “far lower than that of AlphaGeometry2” (lines 311-312) is not clear to me. InternGeometry does pass@256 and uses a larger model (32B instead of 3.3B), so it’s not clear that it uses "far less" computational resources than AlphaGeometry2.**
>
> Thanks for the question. As we responded to question 1, we discussed the computation cost from different perspectives. From various perspectives including **the equivalent number of solutions explored during inference**, **the overall inference steps**, and **the environment execution cost**, InternGeometry is more efficient than AlphaGeometry 2.

---

### Official Review · Reviewer_vjZE · 2025-11-10

**Soundness:** 3
**Presentation:** 3
**Contribution:** 3
**Rating:** 8
**Confidence:** 2

**Summary:**

This work introduces InternGeometry, a LLM agent designed to solve geometry proof problems at the IMO level. By integrating Complexity-Boosted Reinforcement Learning (CBRL) and dynamic memory mechanisms, the model achieves gold-medalist performance on IMO geometry problems, significantly outperforming strong baselines AlphaGeometry2 and SeedGeometry. Notably, InternGeometry only uses 13K training samples, two orders of magnitude less than AlphaGeometry and SeedGeometry. The experiments are thorough, demonstrating the effectiveness in data efficiency and long-range reasoning.

**Strengths:**

1. Propose the first LLM agent for IMO-level geometry proving, avoiding the use of specialist models.
2. Propose a dynamic memory mechanism and rejection sampling strategy, enabling up to 200-step interactive reasoning and guiding diverse explorations in interactions.
3. Solid experiments well justify that InternGeometry outperforms current SOTA models, with exceptional data efficiency. Comprehensive ablation studies validate the necessity of key components like long-range interactions, CBRL, and dynamic memory. The case study justifies the model's creative construction capabilities.

**Weaknesses:**

1. The title is not clear. Since the manuscript focuses on developing plane geometry prover, the title should contain such information.
2. Equation 5 extends beyond the page margin.
3. Experiments on more datasets (e.g., JGEX-AG-231 proposed in AlphaGeometry) and other LLMs (other than InternThinker) can further demonstrate the generalization ability of InternGeometry.
4. Considering that the interactive reasoning requires many steps, analyzing and comparing the computational resources of InternGeometry, AlphaGeometry, and SeedGeometry during reasoning is necessary.

**Questions:**

1. Line 40-42: Add references to justify LLM agents can obtain medalist-level performance on IMO-level problems.

---

> ### Author Response · Authors · 2025-11-23
> **1. The title is not clear. Since the manuscript focuses on developing plane geometry prover, the title should contain such information.**
>
> We thank the Reviewer vjZE for your positive feedback. We are happy to address the reviewer’s concerns and will incorporate all feedback to improve the paper.
>
> Thank you for this helpful suggestion. We agree that the title should more clearly reflect the manuscript’s focus on developing geometry prover. Because the title cannot be changed while the manuscript is under review, we will revise the title when the title revision is available.

---

> ### Author Response · Authors · 2025-11-23
> **2. Equation 5 extends beyond the page margin.**
>
> We have revised Eq. (5) in the paper to ensure that it does not extend beyond the page margins. Please refer to the revised paper for details. We sincerely thank the reviewer again for pointing out this issue.

---

> ### Author Response · Authors · 2025-11-23
> **3. Experiments on more datasets (e.g., JGEX-AG-231 proposed in AlphaGeometry) and other LLMs (other than InternThinker) can further demonstrate the generalization ability of InternGeometry.**
>
> We first evaluate our method on JGEX-AG-231, a diverse geometry benchmark on which most problems are relatively easy. Because AlphaGeometry's performance on this dataset is nearly saturated (228/231), we report pass@32 for InternGeometry to better highlight efficiency. InternGeometry attains 229/231 with pass@32, outperforming AlphaGeometry's 228/231, which was achieved under a substantially heavier search configuration (beam size = 512, maximum iterations = 16, branching factor = 32). Furthermore, with pass@256, InternGeometry solves 230/231.
>
> |      Model     |                              Budget                             | JGEX-AG-231 |
> |:--------------:|:---------------------------------------------------------------:|:-----------:|
> | AlphaGeometry  | beam size = 512，maximum iterations = 16，branching factor = 32 | 228         |
> | InternGeometry | Pass@32                                                         | 229         |
> | InternGeometry | Pass@256                                                        | 230         |
>
>
> We also provide a detailed error analysis. The only unsolved case involves rare 'trisect' constructions that appear only once in the entire JGEX-AG-231 dataset. This construction is not covered by our data-generation pipeline and was never encountered by InternGeometry during training. A preliminary inspection suggests that addressing this question may require specialized geometric techniques beyond the capabilities of the current DDAR system, which we plan to incorporate in future work.
>
> a b c = triangle a b c; d e = trisect d e b a c; f g = trisect f g c b a; h i = trisect h i a c b; j = intersection_ll j b f c i; k = intersection_ll k a e c h; l = intersection_ll l a d b g ? cong j l j k
>
> In addition, we evaluate our method using other backbone models to assess how backbone capacity influences geometric reasoning and to examine the generality of our CBRL framework across different LLMs. Specifically, we report results for QWQ-32B and Intern-s1-mini (8B) on the IMO-50 benchmark under a consistent evaluation protocol (Pass@256). All of these models are available on Hugging Face.
>
> | Backbone            | IMO-50 (Pass@256) |
> |---------------------|-------------------|
> | InternThinker-32B   | 44/50             |
> | QWQ-32B             | 44/50             |
> | Intern-s1-mini (8B) | 40/50             |
>
> These results yield two key observations. First, the 32B backbones substantially outperform the 8B mini backbone (44/50 vs. 40/50), indicating that larger models offer stronger geometric reasoning capabilities in our setting. Second, QWQ-32B attains the same score as InternThinker-32B, suggesting that our CBRL method is not tied to particular backbone and generalizes well across different backbones.

---

> ### Author Response · Authors · 2025-11-23
> **4. Considering that the interactive reasoning requires many steps, analyzing and comparing the computational resources of InternGeometry, AlphaGeometry, and SeedGeometry during reasoning is necessary.**
>
> Thanks for the suggestion. Because neither AlphaGeometry 2 nor SeedGeometry has released open-source code or models, we cannot perform a direct, controlled comparison. Therefore, we rely on the configurations reported in their respective papers to provide a rough estimate and comparison. Since the SeedGeometry paper does not detail its inference budget, we primarily base our equivalent estimates on AlphaGeometry 2.
>
> According to the AlphaGeometry 2 paper, the method uses a Shared Knowledge Ensemble of Search Trees (SKEST) that integrates classical beam search with several tree-search variants (e.g., predicting multiple auxiliary points at each node) and runs multiple trees with different search configurations and models in parallel. Consequently, the total inference budget scales with the number of configurations × the number of instantiated trees × the number of models. The reported optimal single-tree configuration uses a beam size of 128, a branching number (samples) of 32, and a beam depth of 4. However, within SKEST there are configurations that incur higher budgets, such as beam size 64 with depth 10 or beam size 512 with depth 4. The AlphaGeometry 2 model size is 3.3B parameters.
>
> We compare inference efficiency from four perspectives: the equivalent number of solutions explored during inference, the overall inference steps, the environment execution cost, and the overall computation cost.
>
> 1. **Equivalent number of solutions explored**. For a single beam search, the equivalent number of explored solutions can be approximated as beam size × branching number. Under the optimal single-tree configuration, this is 128 × 32 = 4,096, and it can be larger for other configurations (e.g., beam size 512). The total number further scales with the number of configurations × the number of instantiated trees × the number of models. By contrast, InternGeometry’s best-of-K inference explores only K solutions—256 in our experiments—so InternGeometry is more solution-efficient.
>
> 2. **Inference steps**. The optimal beam-tree configuration results in 128 × 32 × 4 = 16,384 steps. Other configurations, such as beam size 64 and depth 10 (64 × 32 × 10 = 20,480) or beam size 512 and depth 4 (512 × 32 × 4 = 65,536), can require more steps. InternGeometry uses a simple pass@256 parallel-inference setting with up to 200 turns of agentic interaction per pass, totaling 51,200 steps. Overall, the per-tree inference budget of AlphaGeometry 2 is on the same order as InternGeometry. However, the total inference cost for AlphaGeometry 2 scales with the number of configurations × the number of instantiated trees × the number of models, making its overall inference steps larger than InternGeometry’s.
>
> 3. **Environment execution cost**. Each step in both AlphaGeometry 2 and InternGeometry is executed in an engine. Because InternGeometry’s total number of steps is smaller than the full SKEST budget of AlphaGeometry 2, it yields fewer total executions. Furthermore, each AlphaGeometry 2 step requires the DDAR system to attempt solving the entire problem, whereas each InternGeometry step either adds an auxiliary construction or attempts to prove a subgoal—operations that are less expensive than attempting to solve the whole problem.
>
> 4. **Overall computation cost**. We acknowledge that InternGeometry’s reasoning style and larger model size increase computation. Each InternGeometry step involves natural-language reasoning, resulting in more tokens and higher compute: for IMO-50, the average number of output tokens per trajectory is 89.6K. The InternGeometry model (32B) is also larger than AlphaGeometry 2 (3.3B). Due to the unknown total cost of AlphaGeometry 2, a direct comparison is difficult. However, we emphasize that the increased computation from deeper reasoning and larger models should not be viewed as a drawback. Instead, it represents a feasible new scaling dimension—alongside training data size and the number of searched solutions—one that aligns more naturally with LLM-based approaches than with expert-model paradigm.
>
> We have also added the discussion in Appendix G.

---

> ### Author Response · Authors · 2025-11-23
> **5. Line 40-42: Add references to justify LLM agents can obtain medalist-level performance on IMO-level problems.**
>
> Thank you for the helpful suggestion. We have updated the Introduction to include citations supporting the claim that LLM agents can reach medalist-level performance on IMO-style problems to better ground the introduction.
>
> 1. Yichen Huang and Lin F. Yang. Winning Gold at IMO 2025 with a Model-Agnostic Verification-and-Refinement Pipeline.
>
> 2. Thang Luong and Edward Lockhart. Advanced version of Gemini with Deep Think officially achieves gold-medal standard at the International Mathematical Olympiad.

---

### Meta-Review · Area_Chair_P5GX · 2026-01-02

**Summary:**

This paper introduces InternGeometry, an LLM-based agent for solving IMO-level geometry problems. The key contributions include: (1) a dynamic memory mechanism enabling up to 200-step interactive reasoning with a symbolic engine, (2) Complexity-Boosted Reinforcement Learning that gradually increases problem difficulty during training, and (3) achieving state-of-the-art performance (44/50 on IMO geometry problems) using only 13K training example (roughly 0.005% of the data used by AlphaGeometry 2). Reviewers raised concerns about computational cost comparisons, clarity of the CBRL algorithm, writing quality, missing details about the base model, and generalization experiments.

**Reviewer Concerns:**

- Computational cost comparison: Authors provided detailed analysis across four dimensions (solutions explored, inference steps, environment execution cost, overall computation), showing InternGeometry is competitive or more efficient than AlphaGeometry 2's SKEST configuration.
- Generalization: Added experiments on JGEX-AG-231 (229/231 with pass@32) and tested multiple backbones (QWQ-32B achieves same 44/50; 8B model achieves 40/50).
- InternThinker-32B clarification: Identified as OREAL-32B (Qwen2.5-32B trained with outcome reward RL), available on HuggingFace.
- CBRL algorithm details: Expanded in Appendix D with pseudocode and theoretical justification.
- Pass@1 reporting: Distribution provided across difficulty intervals.
- Cold-start data curation: Process explained in detail.


- Writing quality issues remain (title cannot be changed during review).
- Some reviewers noted the presentation could still be clearer in places.

**Reviewer Scores:**

- Reviewer vjZE (8): Would likely maintain score; concerns adequately addressed.
- Reviewer YNAr (6): Would likely increase to 7; computational and generalization concerns addressed, though writing concerns partially remain.
- Reviewer 2RDT (8): Would likely maintain score; CBRL clarification and cost analysis were satisfactory.
- Reviewer 5LSi (6): Would likely increase to 7; Pass@1, failure analysis, and compute requirements addressed.

---

### Decision · Program_Chairs · 2026-01-26

Accept (Poster)